# GoIRL: Graph-Oriented Inverse Reinforcement Learning for Multimodal Trajectory Prediction

**Muleilan Pei**[1] **Shaoshuai Shi**[2] **Lu Zhang**[3] **Peiliang Li**[3] **Shaojie Shen**[1]

## Abstract

Trajectory prediction for surrounding agents is a challenging task in autonomous driving due to its inherent uncertainty and underlying multimodality. Unlike prevailing data-driven methods that primarily rely on supervised learning, in this paper, we introduce a novel **G**raph-**o**riented **I**nverse **R**einforcement **L**earning (GoIRL) framework, which is an IRL-based predictor equipped with vectorized context representations. We develop a feature adaptor to effectively aggregate lane-graph features into grid space, enabling seamless integration with the maximum entropy IRL paradigm to infer the reward distribution and obtain the policy that can be sampled to induce multiple plausible plans. Furthermore, conditioned on the sampled plans, we implement a hierarchical parameterized trajectory generator with a refinement module to enhance prediction accuracy and a probability fusion strategy to boost prediction confidence. Extensive experimental results showcase our approach not only achieves state-of-the-art performance on the large-scale Argoverse & nuScenes motion forecasting benchmarks but also exhibits superior generalization abilities compared to existing supervised models.

## 1. Introduction

Trajectory prediction for surrounding traffic participants plays a pivotal role in modern autonomous driving systems, serving as a crucial bridging module that connects upstream perception to downstream planning. Accurately anticipating the future behaviors of nearby agents is paramount to ensure

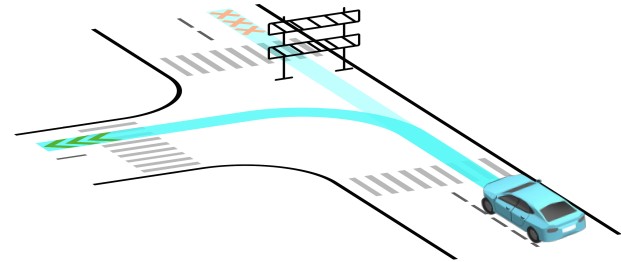

Figure 1. A motivating example of the covariate shift issue in trajectory prediction. In a T-junction driving scenario, the target agent has two potential future trajectories: going straight and turning left. During the data collection process, the ground-truth trajectory is labeled as going straight. However, during testing, the presence of a tentative barricade set up in the intersection renders the original prediction unreasonable. Hence, the predictor must adapt to this change and provide an updated prediction, considering only the option of turning left in this case.

autonomous vehicles can make safe, efficient, and judicious decisions in intricate urban scenarios. However, the task of motion forecasting remains challenging because of its inherent uncertainty and underlying multimodality: an agent can exhibit multiple plausible future trajectories given its past tracks and the available scene information (Song et al., 2021; Shi et al., 2022; Huang et al., 2023a).

Learning-based approaches have emerged as the dominant paradigm due to their powerful representation capabilities to encode the historical motion profiles of agents and the topological and semantic information of environments. In general, data-driven trajectory prediction can be seen as imitation learning from human drivers, i.e., neural networks aim at learning human-like driving behaviors or strategies by leveraging a large amount of recorded driving data as expert demonstrations. This typical robotic task can be approached mainly in two ways: behavior cloning (BC) and inverse reinforcement learning (IRL). Most advanced methods in motion forecasting predominantly employ the BC framework, which involves directly learning the distribution of trajectories from real-world datasets in a supervised manner. In contrast, the IRL architecture models the agent's behavior as a sequential decision-making process (Osa et al., 2018) and aims to infer the underlying reward function that is con-

---

[1]Department of Electronic and Computer Engineering, Hong Kong University of Science and Technology, Hong Kong, China [2]Voyager Research, Didi Chuxing, China [3]Zhuoyu Technology Co., Ltd., Shenzhen, China. Correspondence to: Shaoshuai Shi <shaoshuaics@gmail.com>.

*Proceedings of the 42$^{nd}$ International Conference on Machine Learning*, Vancouver, Canada. PMLR 267, 2025. Copyright 2025 by the author(s).

sidered the most parsimonious and robust representation of the expert demonstrations (Ng & Russell, 2000).

Despite achieving impressive performance in trajectory prediction benchmarks, current supervised models still face challenges concerning generalization or domain adaptation, whereas IRL offers a promising pathway to alleviate them. Specifically, when the situation encountered is significantly out-of-distribution of training demonstrations (referred to as covariate shift in BC (Chen et al., 2023; Yao et al., 2024)), there is a risk of severe compounding error, yielding unreasonable forecasts. For instance, consider a driving scenario where a temporary roadblock is set up in an intersection, causing a previously free space to become an undrivable area, as depicted in Figure 1. Existing supervised models can hardly react to such changes, resulting in unaltered predictions since the novel scene substantially differs from the training time. Besides, few approaches adequately consider drivable information while also having the capacity to effectively utilize this essential attribute, as the planning module does. However, IRL can handle such situations in principle, due to its reward-driven paradigm and learning from interaction property rather than directly fitting data distributions (see Figure 5). Another critical concern associated with the supervised manner is the modality collapse issue. As only one ground-truth future trajectory is provided as supervision, the predictor has to generate diverse plausible predictions via learning one-to-many mappings (Ridel et al., 2020). In contrast, the IRL framework holds the potential to address uncertainty by integrating the maximum entropy (MaxEnt) principle (Ziebart et al., 2008). The MaxEnt IRL approach intends to acquire the reward distribution with the highest entropy (Kretzschmar et al., 2016), which can better capture the intrinsic multimodality of demonstrations. Moreover, the learned reward, as an interpretable intermediate representation (Zeng et al., 2019), can also benefit downstream decision-making and behavior planning in highly complex and interactive scenarios (Huang et al., 2023b).

In light of its superior properties, the MaxEnt IRL framework has garnered significant attention in recent research (Finn et al., 2016a; Zhang et al., 2018). However, most traditional IRL algorithms typically operate efficiently in grid-like environments, which prompts existing work to adopt rasterized context representations, rendering scene elements into bird's-eye-view (BEV) images as input (Kitani et al., 2012; Guo et al., 2022). Consequently, the performance of IRL-based predictors is hampered by scene information loss and inefficient feature extraction when compared to current supervised models with vectorized representations (Gao et al., 2020; Liang et al., 2020). To bridge this gap, we present a feature adaptor capable of aggregating lane-graph features into grid space, thereby empowering IRL-based predictors with vectorized context information. Another bottleneck lies in the computational expense of running IRL,

as it necessitates a forward RL-style procedure in the inner loop (Ho & Ermon, 2016), making it challenging to scale up for larger state spaces. To mitigate this concern, we propose a hierarchical architecture that generates predicted trajectories in a coarse-to-fine manner. Concretely, on the initial coarse scale, we generate multimodal trajectories conditioned on diverse paths or policies drawn from the learned reward distribution using Markov chain Monte Carlo (MCMC) sampling (Hastings, 1970). Additionally, our approach employs a Bézier curve-based parameterization method that recurrently forecasts control points to represent the trajectory, ensuring numerical stability and smoothness. Furthermore, the MCMC-induced distribution is also fused to constitute the final hybrid probability for each modality, thereby enhancing prediction confidence. On the subsequent fine scale, we introduce a trajectory refinement module that leverages both observed and predicted trajectories to retrieve fine-grained local context features, enabling more consistent and precise trajectory forecasts.

Overall, the main contributions of this paper can be summarized as follows: (1) We present a novel **G**raph-**o**riented **I**nverse **R**einforcement **L**earning (GoIRL) framework for the multimodal trajectory prediction task, which, to the best of our knowledge, is the first to integrate the MaxEnt IRL paradigm with vectorized context representation through our proposed feature adaptor. (2) We introduce a hierarchical parameterized trajectory generator for improving prediction accuracy and an MCMC-augmented probability fusion for boosting prediction confidence. (3) Our approach achieves state-of-the-art performance on two large-scale motion forecasting benchmarks, namely Argoverse (Chang et al., 2019) and nuScenes (Caesar et al., 2020). Further, it demonstrates superior generalization abilities compared to existing supervised models in handling drivable area changes.

## 2. Related Work

### 2.1. Scene Context Representation

Scene context, including high-definition (HD) maps, offers valuable information that significantly contributes to trajectory prediction. Early works often rasterize the environment into a BEV image, with distinct colors representing different types of input information. Scene features are subsequently extracted using convolutional neural networks (CNNs) and pooling layers (Cui et al., 2019). Nevertheless, these rasterization-based methods have limitations in capturing long-range spatiotemporal relationships owing to the lack of detailed geometric information from road maps (Nayakanti et al., 2022). Therefore, a more effective and powerful alternative, known as vectorized (Gao et al., 2020) or graph-based (Liang et al., 2020) representation, is then proposed, which can provide abundant geometric and semantic information, such as map topology, lane connectivity,

and correlation relationships among neighborhood agents. Building upon these vectorized data, graph neural networks (GNNs) and Transformer-based architectures have been extensively explored for feature extraction and fusion (Zeng et al., 2021; Zhou et al., 2023). This paradigm has gained increasing popularity in current prediction models due to its remarkable enhancement to overall prediction performance. Consequently, in our framework, we also adopt the prevalent vectorized representation to encode the scene context and employ graph-based models to extract lane-graph features.

## 2.2. Multimodal Trajectory Prediction

Multimodal trajectory prediction necessitates generating diverse, socially acceptable, and interpretable future trajectories accompanied by corresponding confidence scores (Huang et al., 2023a). One practical approach is to utilize stochastic techniques, such as generative adversarial network (GAN) (Gupta et al., 2018) or conditional variational autoencoder (CVAE) (Ivanovic et al., 2020), to generate multiple possible forecasts. However, sampling from a latent distribution is uncontrollable during inference, prompting recent efforts to focus on anchor-based methods instead. Predefined anchors, including goal points (Gu et al., 2021), reference lanes (Song et al., 2021), or candidate paths (Afshar et al., 2024), can act as mode-specific priors to facilitate forecasting diversity. Nevertheless, the accuracy of predictions heavily relies on the quality of these anchors. By contrast, our work leverages MaxEnt IRL to infer the intrinsic multimodal distribution of rewards by maximizing entropy. Multiple plausible policies or plans can then be sampled from the learned reward distribution, generating diverse trajectories. Furthermore, the probability of each modality is commonly obtained through classification with hard assignments using displacement regression (Ye et al., 2021). Yet, we contend that the confidence scores obtained in this manner are inadequate to characterize uncertainty comprehensively and thus propose to augment the confidence scores by fusing the MCMC-induced distribution.

## 2.3. Inverse Reinforcement Learning

Inverse Reinforcement Learning (IRL), a prominent branch of imitation learning, has been extensively investigated in the domains of robotic manipulation (Finn et al., 2016b), navigation (Fu et al., 2018), and control (Yu et al., 2019; Osa et al., 2018). It aims to recover the underlying reward distribution from expert demonstrations, enabling the derivation of policies consistent with human behaviors. Supposing that surrounding agents are mostly rational and thus can be modeled as optimal or suboptimal planners, the IRL approach becomes a powerful tool for motion forecasting, functioning as a planning-based predictor. In order to reason about uncertainty in decision sequences, a probabilistic approach known as MaxEnt IRL has been proposed and successfully

applied to infer both future destinations and routes in real-world navigation tasks (Ziebart et al., 2008). Nevertheless, previous IRL-based predictors have primarily been designed for path forecasting (Ziebart et al., 2009), neglecting the consideration of time profiles. To overcome this disparity, recent studies integrate kinematics and environmental context into the IRL scheme (Zhang et al., 2018), presenting a plans-to-trajectories module that can generate continuous trajectories conditioned on grid-based state sequences sampled from the reward distribution (Deo & Trivedi, 2020; Guo et al., 2022). However, existing IRL-based predictors solely utilize rasterized BEV features as input, which limits their prediction performance. In contrast, our work introduces a novel feature adaptor designed to incorporate lane-graph features into the MaxEnt IRL framework, ultimately enhancing prediction accuracy.

# 3. Methodology

## 3.1. Problem Formulation

The multimodal trajectory prediction task entails generating multiple plausible future trajectories for the target agent, each accompanied by corresponding probabilities conditioned on the scene context. More specifically, considering the driving context $\mathcal{C} = \{\mathcal{X}, \mathcal{O}, \mathcal{M}\}$, which consists of the tracking positions of the target agent $\mathcal{X} = [x_{-t_p+1}, \ldots, x_0]$ and its observed neighboring agents $\mathcal{O} = [o_{-t_p+1}, \ldots, o_0]$ spanning the past $t_p$ timestamps, as well as the relevant map information $\mathcal{M}$, the motion predictor aims to infer the probability distribution of future forecasts $P(\hat{\mathcal{Y}}|\mathcal{C})$, where $\hat{\mathcal{Y}} = [\hat{y}_1, \ldots, \hat{y}_{t_f}]$ represents the predicted future trajectory of the target agent over the future $t_f$ timestamps.

To tackle trajectory prediction using IRL-based methods, we decompose the entire task into two conceptual stages: policy inference and trajectory generation. The scene context $\mathcal{C}$ is first aggregated into a coarse 2-D grid space. Then, the IRL framework is characterized as a finite Markov decision process (MDP) with a fixed horizon $\mathcal{H}$, which is defined as a four-tuple $\{\mathcal{S}, \mathcal{A}, \mathcal{T}, \mathcal{R}\}$. Herein, $\mathcal{S}$ represents a set of states comprising all grid cells, while $\mathcal{A}$ represents a set of actions consisting of nine discrete movements, including eight adjacent and diagonal directions, along with one *end* action for the early rollout termination. $\mathcal{T} : \mathcal{S} \times \mathcal{A} \to \mathcal{S}$ denotes the deterministic transition model and $\mathcal{R}$ denotes reward function which is initially unknown. Next, we leverage a stationary policy $\pi(a|s)$ derived from the learned reward distribution to specify a probabilistic mapping from a state $s \in \mathcal{S}$ to a particular action $a \in \mathcal{A}$. Ultimately, the target agent makes forecasts conditioned on grid-based plans sampled from the optimal policy $\pi^*$, where each plan is referred to as a sequence of states given by $\hat{\tau} = [\hat{s}_1, \ldots, \hat{s}_{\mathcal{H}}]$. Consequently, the multimodal future trajectory distribution $P(\hat{\mathcal{Y}}|\mathcal{C})$ can be decomposed by conditioning on plans and

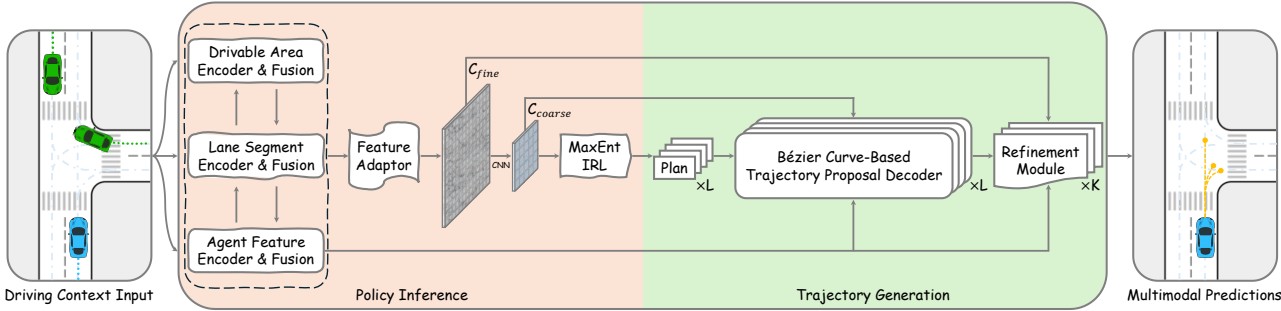

*Figure 2.* Overview of GoIRL, illustrating the generation of multimodal trajectory predictions for the target agent (depicted in blue).

then marginalizing over them:

$$P(\hat{\mathcal{Y}}|\mathcal{C}) = \sum_{\hat{\tau} \in \mathbb{S}(\mathcal{C})} P(\hat{\mathcal{Y}}|\hat{\tau}, \mathcal{C}) P(\hat{\tau}|\mathcal{C}), \qquad (1)$$

where $\mathbb{S}(\mathcal{C})$ denotes the space of plausible plans based on the driving context, which is derived from the policy inference.

### 3.2. Framework Overview

An overview of our proposed framework is presented in Figure 2, which illustrates a two-stage architecture involving policy inference and trajectory generation. In the first stage, we encode the driving context in a vectorized manner and employ a graph-based approach to extract scene features. The lane-graph features are then aggregated into grid space using the proposed feature adaptor and further compressed through CNNs to reduce the state-space dimension. Afterward, the grid-based MaxEnt IRL algorithm is utilized to infer the reward distribution, thereby obtaining the optimal policy that can be sampled to induce multiple plausible plans or traversals over the 2-D grid. In the second stage, we introduce a hierarchical structure for generating multimodal future trajectories conditioned on the sampled plans and multiscale scene embedding features. Concretely, we parameterize predicted trajectories using Bézier curves and recurrently produce control points to represent continuous trajectories as initial proposals. Subsequently, a refinement module is designed to generate location offsets for each clustered proposal by retrieving fine-grained local context embedding features via the complete trajectory, which comprises both the predicted proposal and past observation, in order to facilitate temporal consistency and spatial accuracy.

### 3.3. Graph-Oriented Context Encoder

Considering context rasterization suffers from information loss and inefficient feature extraction, we employ a graph-oriented context encoder to aggregate scene features in a vectorized manner, enabling better capture of complicated topology connectivity and long-range interactions. Specifically, we adopt an agent-centric double-layered graph structure for encoding the driving context. Here, we take both lane segments and drivable areas into account because the drivable areas also offer valuable traffic information. Firstly, we construct a lane graph based on the vectorized map data, where each lane node represents a polyline segment and provides essential geometric and semantic attributes, including the types of connections. Then, the dilated LaneConv operator proposed in LaneGCN (Liang et al., 2020) is applied to extract the lane-graph features. Subsequently, we uniformly sample nodes within drivable areas and build an interconnected spatiotemporal occupancy graph. Afterward, the node features of the drivable area are updated using a PointNet-like network (Qi et al., 2017) with dilated connections. In addition, we leverage a 1-D CNN with a feature pyramid network (FPN) (Lin et al., 2017) to encode the kinematic profiles of all agents in the scene, encompassing historical positions, velocities, time intervals, etc. Finally, a stack of fusion modules with graph attention layers is employed to capture the interactions among the aforementioned vectorized embedding features and update the final fused context features. More details of the encoding process can be found in Appendix A.1.

**Feature Adaptor.** As efficient IRL-based methods typically rely on image- or grid-shaped features as input, we devise a feature adaptor to aggregate the graph-based context features into grid space. To achieve this, we construct a grid map centered around the target agent, with each grid cell uniformly spaced and aligned with drivable nodes. The fused context features within the drivable areas are then assigned to their corresponding cells based on physical locations, while the undrivable cells are padded with zeros. This operation successfully transforms the vectorized features into grid-shaped driving context features, denoted $\mathcal{C}_{fine}$, which can be seamlessly adapted to the IRL framework. Moreover, considering the high-resolution grid space leads to the IRL process being extremely time-consuming, we employ CNNs with strides to downsample the spatial dimension of the original feature map, effectively alleviating the computational burden without losing essential information. Consequently, the coarse-grained context features denoted

**Algorithm 1** Approximate Value Iteration

**Input:** $\mathcal{S}, \mathcal{A}, \mathcal{T}, \mathcal{R}$
**Output:** $\pi(a|s)$
1: $Q(s,a) = 0, V(s) \leftarrow -\infty, \ \forall s \in \mathcal{S}, a \in \mathcal{A}$
2: **for** $i = 1$ **to** $N$ **do**
3: $\quad V(\tilde{s}) \leftarrow \mathcal{R}(\tilde{s}), \ \tilde{s} \in \mathcal{S}_{goal}$
4: $\quad Q(s,a) = \mathcal{R}(s) + V(s'), \ s' = \mathcal{T}(s,a)$
5: $\quad V(s) = \log \left( \sum_a \exp \left( Q(s,a) \right) \right)$
6: **end for**
7: $\pi(a|s) = \exp \left( Q(s,a) - V(s) \right)$

$\mathcal{C}_{coarse}$ are further extracted using $1 \times 1$ CNNs to yield the reward distribution, denoted $\mathcal{R}$. Overall, the reward can be regarded as a nonlinear combination of the driving context throughout the entire encoding process.

### 3.4. MaxEnt IRL-Based Policy Generator

After modeling the reward function as a neural network, we aim to infer the optimal policy that can generate multiple plausible plans within the grid. Following the MaxEnt IRL framework (Ziebart et al., 2008), the probability of a state sequence (or plan) is proportional to the exponential of the total reward accumulated over the planning horizon $\mathcal{H}$:

$$P(\tau) = \frac{1}{Z} \exp \left( \mathcal{R}(\tau) \right) = \frac{1}{Z} \exp \left( \sum_{i=1}^{\mathcal{H}} \mathcal{R}(s_i) \right), \quad (2)$$

where $\tau = [s_1, \ldots, s_{\mathcal{H}}]$ represents any given plan and $Z$ denotes the partition function. Further, we translate the continuous-valued future trajectories from the dataset into discrete state sequences using a uniform quantization technique with a specific resolution, constituting a set of demonstrations denoted as $\mathcal{D} = \{\tau_1, \ldots, \tau_{|\mathcal{D}|}\}$. The objective is to maximize the log-likelihood of the demonstration data $\mathcal{L}_{\mathcal{D}}$ under the MaxEnt distribution. This optimization problem can be solved using gradient-based approaches, and the gradient is given by

$$\nabla \mathcal{L}_{\mathcal{D}} = (\mu_{\mathcal{D}} - \mathbb{E}[\mu]) \nabla \mathcal{R}, \quad (3)$$

where $\mu_{\mathcal{D}}$ represents the average state visitation frequencies (SVFs) from the demonstrations and $\mathbb{E}[\mu]$ refers to the expected SVFs under the policy (Wulfmeier et al., 2017), which can be derived via a forward RL process given the current reward distribution. Here, we leverage the approximate value iteration algorithm (detailed in Algorithm 1) to obtain the MaxEnt policy $\pi(a|s)$ over $N$ update steps with the following expression:

$$\pi(a|s) = \exp \left( Q(s,a) - V(s) \right), \quad (4)$$

where $Q(s,a)$ represents the action-value function and $V(s)$ refers to the state-value function. Note that since the goal

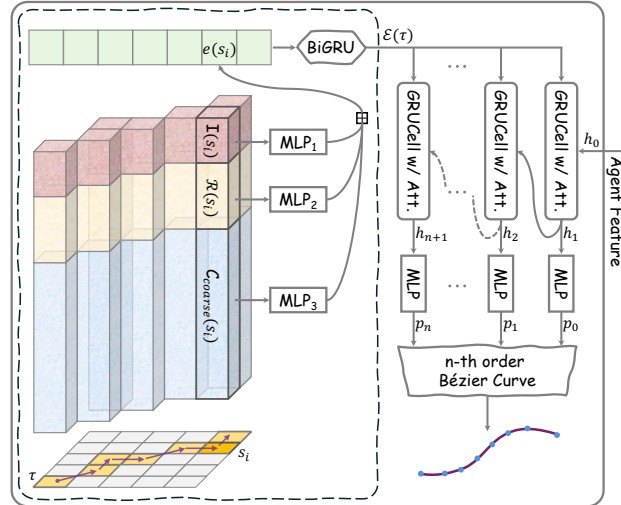

*Figure 3.* Structure of Bézier curve-based trajectory proposal decoder for each sampled plan.

state set $\mathcal{S}_{goal}$ is unknown in the trajectory prediction task, the terminal state distribution is also inferred by the neural networks. Upon convergence of the reward distribution, we can acquire the optimal MaxEnt policy $\pi^*$, which enables the generation of multiple plans over the 2-D grid, serving as trajectory generation priors.

### 3.5. Hierarchical Multimodal Trajectory Decoder

Based on the converged reward model $\mathcal{R}$ and the optimal MaxEnt policy $\pi^*$, along with multiscale driving context features $\mathcal{C}_{coarse}$ and $\mathcal{C}_{fine}$, we present a hierarchical trajectory decoder to generate $K$ multimodal predictions and their corresponding probabilities in a coarse-to-fine manner.

On the initial coarse scale, we generate multiple plausible trajectory proposals conditioned on the sampled plans and associated coarse-grained context features, as illustrated in Figure 3. Specifically, we first employ the learned MaxEnt policy to induce plans over the 2-D grid cells using the Markov chain Monte Carlo (MCMC) sampling strategy. Each plan is then utilized to extract features from the coarse-grained context feature map $\mathcal{C}_{coarse}$ and the reward map $\mathcal{R}$ along the state sequence. For each state (or grid) $s_i$ within a plan $\tau$, we adopt simple multilayer perceptrons (MLPs) to encode its local context features $\mathcal{C}_{coarse}(s_i)$, reward $\mathcal{R}(s_i)$, and grid location coordinates $\mathcal{I}(s_i)$, respectively, and concatenate them as the state feature embedding $e(s_i)$, i.e.,

$$e(s_i) = \left( f_1(\mathcal{C}_{coarse}(s_i)) \boxplus f_2(\mathcal{R}(s_i)) \boxplus f_3(\mathcal{I}(s_i)) \right), \quad (5)$$

where $\boxplus$ represents the concatenation operator, and $f_j, j = 1, 2, 3$ denotes a simple MLP block. The state feature embeddings $e(s_i)$ over the entire plan $\tau$ are further aggregated via a bidirectional gated recurrent unit (BiGRU) encoder, yielding the plan feature $\mathcal{E}(\tau)$. Next, instead of directly

predicting the future trajectory positions, we employ an $n$-th order Bézier curve to represent continuous trajectories. Herein, we adopt a GRU decoder with the soft attention mechanism (Bahdanau et al., 2015) to recurrently generate $n + 1$ control points:

$$p_i = \varphi(h_{i+1}), \tag{6}$$

$$h_{i+1} = \mathcal{G}(\text{Att}(\mathcal{E}(\tau), h_i), h_i), \tag{7}$$

where $p_i, i = 0, \ldots, n$, is the control point for the Bézier curve, $h_i$ is the hidden state of the GRU, and the initial one $h_0$ is the updated target agent feature obtained by aggregating fused context features and interactions among agents through the agent fusion module. $\varphi$ denotes an MLP with a single hidden layer, $\mathcal{G}$ represents the GRU cell, and $\text{Att}(\cdot)$ denotes the soft attention module. Once the control points are generated, the continuous trajectory, denoted $\mathcal{B}(t)$, can be obtained using the following expression:

$$\mathcal{B}(t) = \sum_{i=0}^{n} b_{i,n}(t) p_i, \ 0 \leq t \leq 1, \tag{8}$$

where $b_{i,n}(t) = \binom{n}{i} t^i (1 - t)^{n-i}, i = 0, \ldots, n$, is the $n$-degree Bernstein basis polynomial, and $\binom{n}{i} = \frac{n!}{i!(n-i)!}$ denotes the binomial coefficient. In practice, for a given timestamp $t_s, t_s \in \{1, 2, \ldots, t_f\}$, the position of the predicted trajectory proposal $\hat{\bar{y}}_{t_s}$ can be determined by substituting $t$ with $\frac{t_s}{t_f}$:

$$\hat{\bar{y}}_{t_s} = \mathcal{B}\left(\frac{t_s}{t_f}\right) = \sum_{i=0}^{n} b_{i,n}\left(\frac{t_s}{t_f}\right) p_i. \tag{9}$$

Considering that sampling only $K$ plans is often inefficient and redundant, as a small number of samples tend to produce similar outputs, we oversample $L$ ($L \gg K$) plans in parallel, thereby inducing $L$ continuous valued trajectories to better capture the trajectory distribution. Subsequently, we apply the K-means clustering method to derive $K$ possible future trajectories from the set of $L$ candidates. Moreover, we determine the probabilities by calculating the proportion of trajectories associated with each clustered modality, denoted $\mathcal{P}_{mcmc}$, which can reflect the statistical property of MCMC sampling from the MaxEnt policy.

On the second fine scale, we generate trajectory offsets to refine the $K$ initial trajectory proposals in conjunction with fine-grained local context features $\mathcal{C}_{fine}$. Concretely, we first combine past observations and predicated proposals together to form the complete trajectory for the target agent, i.e., $[\mathcal{X}, \hat{\bar{\mathcal{Y}}}^i], i = 1, \ldots, K$, where $\hat{\bar{\mathcal{Y}}}^i$ denotes the $i$-th predicted trajectory proposal. For each complete trajectory, we gather its nearby fine-grained context features. These embeddings are then aggregated along the entire trajectory using global average pooling and concatenated with the

previous updated agent features, as well as the location embeddings obtained by encoding the complete trajectory locations through an MLP block. The fused features are further fed into MLP blocks with residual connections, which comprises a regression head for producing the predicted trajectory offsets, denoted $\Delta \hat{\mathcal{Y}}^i$, and a classification head followed by a softmax function for generating the probabilities, denoted $\mathcal{P}_{cls}$.

Eventually, the $i$-th predicted trajectory $\hat{\mathcal{Y}}^i$ can be derived by summing the trajectory proposal and offset:

$$\hat{\mathcal{Y}}^i = \hat{\bar{\mathcal{Y}}}^i + \Delta \hat{\mathcal{Y}}^i. \tag{10}$$

Its corresponding fusion probability takes a hybrid form:

$$\mathcal{P}(\hat{\mathcal{Y}}^i) = \frac{\mathcal{P}_{cls}(\hat{\mathcal{Y}}^i) \mathcal{P}_{mcmc}(\hat{\mathcal{Y}}^i)}{\sum_{i=1}^{K} \mathcal{P}_{cls}(\hat{\mathcal{Y}}^i) \mathcal{P}_{mcmc}(\hat{\mathcal{Y}}^i)}. \tag{11}$$

### 3.6. Learning

Given that our proposed framework consists of two stages, the training process necessitates a decomposition into two phases. In the initial MaxEnt IRL stage, our objective is to maximize the log-likelihood $\mathcal{L}_\mathcal{D}$ through stochastic gradient descent to derive the reward model and optimal policy, as explained in Section 3.4. In the second trajectory generation stage, the overall learning objective is composed of both regression loss and classification loss. Specifically, for regression, we apply the Huber loss to the predicted trajectory proposal $\mathcal{L}_{reg}^{\text{P}}$, the refined trajectory $\mathcal{L}_{reg}^{\text{T}}$, and its corresponding goal point $\mathcal{L}_{reg}^{\text{G}}$. The winner-takes-all (WTA) training strategy is employed to mitigate the modality collapse issue, which only considers the best candidate with the minimum displacement error in comparison to the ground truth. As for classification, we adopt the Hinge loss $\mathcal{L}_{cls}$ to distinguish the positive modality from the others, following the approach outlined in (Liang et al., 2020). The total loss $\mathcal{L}$ in the second training stage can be expressed as follows:

$$\mathcal{L} = \mathcal{L}_{reg}^{\text{P}} + \alpha \mathcal{L}_{reg}^{\text{T}} + \beta \mathcal{L}_{reg}^{\text{G}} + \gamma \mathcal{L}_{cls}, \tag{12}$$

where $\alpha$, $\beta$, and $\gamma$ are hyperparameters for balancing each loss component. In practice, we set $\alpha = \beta = 1$ and $\gamma = 3$. More details can be found in Appendix A.2.

## 4. Experiments and Results

### 4.1. Experiment Setup

**Real-World Datasets.** We train and evaluate the proposed approach on two large-scale public motion forecasting datasets: Argoverse (Chang et al., 2019) and nuScenes (Caesar et al., 2020). Both datasets provide trajectory sequences collected from real-world urban driving scenarios,

along with HD maps that encompass rich geometric and semantic information. Specifically, the Argoverse dataset comprises 205,942 training, 39,472 validation, and 78,143 testing sequences. Each sequence spans 5 seconds and is sampled at 10 Hz. The task involves forecasting the subsequent 3-second trajectories based on the preceding 2 seconds of observations. As for the nuScenes dataset, it contains 32,186 training, 8,560 validation, and 9,041 testing sequences. Each sequence is 8 seconds long and sampled at 2 Hz. The goal is to predict the future 6-second trajectories given the past 2 seconds of observations.

**Evaluation Metrics.** We evaluate the performance of multimodal trajectory prediction using the widely accepted metrics, including miss rate ($MR_K$), minimum average displacement error ($minADE_K$), minimum final displacement error ($minFDE_K$), and the Brier minimum final displacement error (brier-$minFDE_K$) specific to the Argoverse benchmark. Concretely, the $MR_K$ measures the proportion of scenarios where none of the $K$ predicted endpoints fall within a 2.0-meter range of the ground truth. The $minADE_K$ calculates the average pointwise $\ell_2$ distance between the best forecast among the $K$ candidates and the ground truth, while the $minFDE_K$ solely focuses on the endpoint error. Furthermore, the brier-$minFDE_K$ incorporates prediction confidence by adding the Brier score $(1.0 - \mathcal{P})^2$ to $minFDE_K$, where $\mathcal{P}$ corresponds to the probability of the best forecast.

**Implementation Details.** We employ the target-centric coordinate system, where all context instances and sequences are normalized to the current state of the target agent through translation and rotation operations. Note that no augmentation techniques are applied in our method. Moreover, the spatial dimension of the grid space is initially defined as $100 \times 100$ and then downsampled to $25 \times 25$ for the MDP. The planning horizon $\mathcal{H}$ is configured as 25. Additionally, during the training phase, we set the degree of the Bézier curve $n = 5$, the oversample number $L = 600$, and the mode number $K = 6$ for Argoverse and $K = 10$ for nuScenes. The first-stage model size amounts to 2.79M parameters, while the second-stage model consists of 0.85M parameters. We train our model on eight GPUs using the AdamW optimizer with a batch size of 256.

## 4.2. Quantitative Results

**Comparison with State-of-the-Art.** We compare our approach with state-of-the-art methods on both Argoverse and nuScenes motion forecasting benchmarks. Table 1 shows quantitative results of the single-model performance on the Argoverse test split. The official ranking metric is shaded in gray, and the best results are indicated in bold. All metrics follow a lower-the-better criterion. To the best of our knowledge, no IRL-based predictor is publicly available on the Argoverse leaderboard. Thus, we compare our GoIRL

Table 1. Single-model results on the **Argoverse** motion forecasting benchmark, ranked by the official metric brier-$minFDE_6$.

| Method | brier-$minFDE_6$ | Brier score | $minFDE_6$ | $minADE_6$ | $MR_6$ |
|---|---|---|---|---|---|
| LaneRCNN (Zeng et al., 2021) | 2.1470 | 0.6944 | 1.4526 | 0.9038 | 0.1232 |
| LaneGCN (Liang et al., 2020) | 2.0585 | 0.6945 | 1.3640 | 0.8679 | 0.1634 |
| DSP (Zhang et al., 2022) | 1.8584 | 0.6398 | 1.2186 | 0.8194 | 0.1303 |
| HeteroGCN (Gao et al., 2023) | 1.8399 | 0.6521 | 1.1878 | 0.8173 | 0.1236 |
| GoIRL (Ours) | **1.7957** | **0.6232** | **1.1725** | **0.8087** | **0.1204** |

Table 2. Ensemble-model results on the **Argoverse** motion forecasting benchmark, ranked by the official metric brier-$minFDE_6$.

| Method | brier-$minFDE_6$ | Brier score | $minFDE_6$ | $minADE_6$ | $MR_6$ |
|---|---|---|---|---|---|
| MultiPath++ (Varadarajan et al., 2022) | 1.7932 | 0.5788 | 1.2144 | 0.7897 | 0.1324 |
| HeteroGCN (Gao et al., 2023) | 1.7512 | 0.5910 | 1.1602 | 0.7890 | 0.1168 |
| SIMPL (Zhang et al., 2024) | 1.7469 | 0.5924 | 1.1545 | 0.7693 | 0.1165 |
| Wayformer (Nayakanti et al., 2022) | 1.7408 | 0.5792 | 1.1616 | **0.7676** | 0.1186 |
| GoIRL (Ours) | **1.6947** | **0.5684** | **1.1263** | 0.7828 | **0.1102** |

Table 3. Quantitative results on the **nuScenes** prediction benchmark, sorted by the official ranking metric $minADE_5$.

| Method | $minADE_5$ | $minADE_{10}$ | $minFDE_1$ | $MR_5$ | $MR_{10}$ |
|---|---|---|---|---|---|
| P2T (Deo & Trivedi, 2020) | 1.45 | 1.16 | 10.5 | 0.64 | 0.46 |
| PGP (Deo et al., 2021) | 1.27 | 0.94 | 7.17 | 0.52 | 0.34 |
| MacFormer (Feng et al., 2023) | 1.21 | 0.89 | 7.50 | 0.57 | 0.33 |
| Goal-LBP (Yao et al., 2023) | 1.02 | 0.93 | 9.20 | 0.32 | 0.27 |
| GoIRL (Ours) | **0.86** | **0.75** | **6.53** | **0.31** | **0.21** |

framework against several representative supervised models with graph-based context encoders. It can be seen that our proposed method achieves the best results across all metrics. In particular, GoIRL surpasses the baseline model DSP (Zhang et al., 2022) by a large margin, highlighting the effectiveness of the IRL architecture in enhancing trajectory predictions.

Furthermore, we adopt an ensemble strategy (Zhang et al., 2024) to boost prediction performance, enabling a comprehensive comparison against other supervised ensemble methods. As shown in Table 2, GoIRL continues to demonstrate highly competitive performance compared to state-of-the-art supervised models like the strong baseline Wayformer (Nayakanti et al., 2022). We also report the results on the nuScenes prediction leaderboard in Table 3 to exhibit the long-term prediction performance of our proposed framework. As presented, GoIRL achieves top-ranked performance on this benchmark, remarkably outperforming all other approaches across all evaluation metrics, including the IRL-based baseline method P2T (Deo & Trivedi, 2020) that represents the driving context in a rasterized manner. This outcome illustrates that by leveraging the feature adaptor to effectively incorporate the lane-graph features, we can significantly improve the upper bound of performance for IRL-based predictors. This advancement holds great potential to empower multimodal trajectory prediction tasks.

*Table 4.* Ablation studies on the Argoverse test split.

| Refine | Bézier curve | Recurrent strategy | $\mathcal{P}_{mcmc}$ | brier-minFDE$_6$ | Brier score | minFDE$_6$ |
|:---:|:---:|:---:|:---:|:---:|:---:|:---:|
| ✗ | ✓ | ✓ | ✓ | 1.8280 | 0.6468 | 1.1812 |
| ✓ | ✗ | ✓ | ✓ | 1.7993 | 0.6242 | 1.1751 |
| ✓ | ✓ | ✗ | ✓ | 1.8000 | 0.6236 | 1.1764 |
| ✓ | ✓ | ✓ | ✗ | 1.8039 | 0.6310 | 1.1729 |
| ✓ | ✓ | ✓ | ✓ | **1.7957** | **0.6232** | **1.1725** |

**Ablation Study.** We conduct thorough ablation studies on the Argoverse test split to investigate the functionality of several key components of the trajectory decoder. Each of them is ablated from the complete pipeline to quantify its individual impact on the final performance, as presented in Table 4. The results reveal that the refinement module makes a substantial contribution to prediction accuracy, underscoring that the hierarchical architecture with effective aggregation of fine-grained features can notably improve location precision. In addition, both the Bézier curve-based trajectory parameterization and the recurrent strategy for point generation bring benefits to distance-aware metrics while also ensuring kinematic feasibility and temporal consistency. Furthermore, the comparison of the Brier score metric clearly demonstrates that the probability fusion with the MCMC-induced distribution $\mathcal{P}_{mcmc}$ leads to remarkable enhancements to the predictive confidence score, thereby emphasizing its efficacy in capturing the multimodal distribution of future trajectories. In summary, all components within our proposed framework play vital roles in attaining superior prediction performance.

### 4.3. Qualitative Results

We showcase some visualizations of our proposed method under diverse traffic scenarios from the Argoverse validation set, as depicted in Figure 4. From the qualitative results, it is evident that our approach successfully anticipates accurate and feasible multimodal future trajectories that conform to the scene layout. Additionally, we apply our model to the Argoverse tracking dataset without any further fine-tuning, as detailed in Appendix C.1. The consecutive trajectory prediction results are available in the supplementary video[1].

**IRL-based v.s. BC-based predictor.** We conduct an investigation into the generalization and domain adaptation abilities of our GoIRL framework with respect to the drivable area attribute. To achieve this, we modify the drivable attribute of a traffic scenario within the original dataset, transforming a previously designated free space into an undrivable area, as demonstrated in Figure 5. Our primary objective is to evaluate the capability of predictors in handling such distribution-shifting cases. Unfortunately, existing methods often fall short of adequately reacting to such

---

[1]https://youtu.be/MPECgueGRaQ

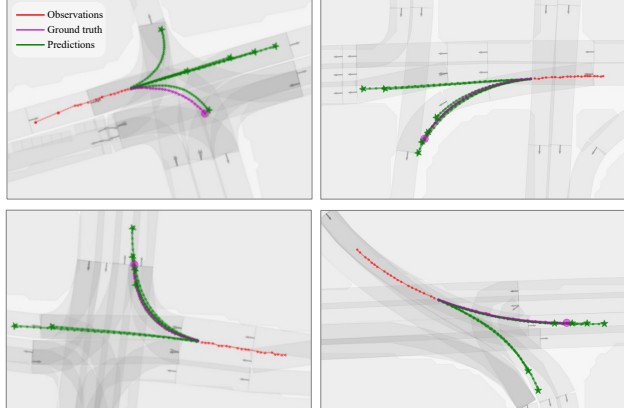

*Figure 4.* Visualizations of GoIRL on the Argoverse validation set. The historical trajectory, ground-truth future trajectory, and multimodal predictions are depicted in red, magenta, and green, respectively. Other traffic participants have been excluded to emphasize the predictions for the target agent.

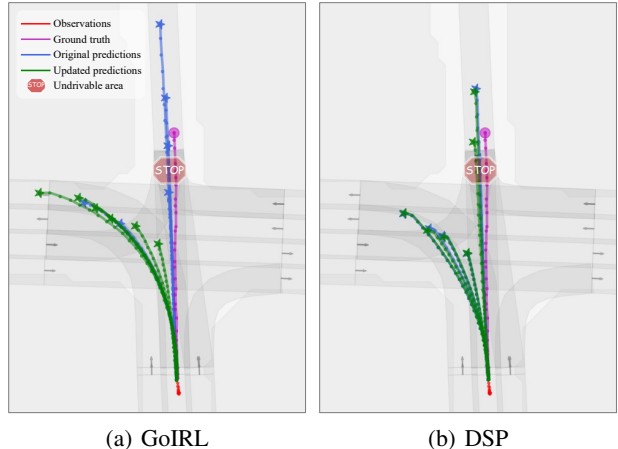

(a) GoIRL          (b) DSP

*Figure 5.* Qualitative comparisons between GoIRL (left) and DSP (right) in handling drivable area changes. The original predictions are depicted in blue, while the updated trajectories are in green. The undrivable area is indicated with a "STOP" sign symbolizing the transformed region.

changes due to the lack of consideration for drivable information, resulting in irrational predictions or even potential crashes. Hence, we compare our approach against DSP (Zhang et al., 2022), which, to the best of our knowledge, is the only supervised model that incorporates drivable information. As shown in Figure 5(a), GoIRL can effectively adapt to changes in the environment and generate reasonable predictions, thanks to the inherent interaction learning property of the IRL paradigm. In contrast, Figure 5(b) showcases that the DSP model fails to respond appropriately, thereby highlighting the superior generalization ability of IRL-based predictors in addressing such domain bias issues. More qualitative results can be found in Appendix C.2.

# 5. Conclusion

In this paper, we introduce GoIRL, a graph-oriented inverse reinforcement learning framework for multimodal trajectory prediction tasks. To the best of our knowledge, GoIRL is the first motion prediction scheme that integrates the MaxEnt IRL paradigm with graph-based context representations through the proposed feature adaptor, facilitating effective scene feature extraction and aggregation. Moreover, the hierarchical parameterized trajectory generator remarkably enhances prediction accuracy, while the MCMC-augmented probability fusion strategy amplifies prediction confidence. Experimental results on large-scale motion forecasting benchmarks demonstrate that GoIRL excels in generating scene-compliant multimodal future trajectories and outperforms the current state-of-the-art methods. The advantages of the IRL paradigm also empower GoIRL with exceptional generalization and domain adaptation capabilities, allowing it to effectively address distribution-shifting challenges, such as drivable area changes, when compared to existing supervised models. Our work underscores the effectiveness of IRL-based motion predictors and provides a promising baseline for further investigations. Future work will concentrate on extending the IRL paradigm to encompass joint multi-agent trajectory forecasting.

## Acknowledgment

This work was supported in part by the Hong Kong Ph.D. Fellowship Scheme, and in part by the HKUST-DJI Joint Innovation Laboratory. The authors thank Xiaogang Jia, Xu Liu, Zhaojin Huang, Wenchao Ding, and Haoran Song for their insightful discussions.

## Impact Statement

This paper presents work whose goal is to advance the field of Machine Learning. There are many potential societal consequences of our work, none of which we feel must be specifically highlighted here.

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

# A. Additional Implementation Details

In this section, we provide a more detailed description of the context encoder, feature fusion, and adaptation, along with the specific formulations of the various training losses.

## A.1. Graph-Oriented Context Encoder

Our graph-oriented context encoder follows a structure similar to the one used in DSP (Zhang et al., 2022), which adopts an agent-centric, double-layered, graph-based representation. This encoder consists of three components: an agent feature encoder, a lane segment encoder, and a drivable area encoder.

**Agent Feature Encoder.** Given the observed past timestamps $t_p$, we define the agent trackings as $A \in \mathbb{R}^{N_a \times t_p \times F_a'}$, where $N_a$ is the number of agents in the scene, and $F_a'$ encompasses kinematic attributes such as positions, velocities, time intervals, and binary masks. To be specific, the agents include the target agent $\mathcal{X}$ and surrounding agents $\mathcal{O}$. Next, We utilize a 1-D CNN with a feature pyramid network (FPN), as implemented in (Liang et al., 2020), to capture multi-scale features. This results in the agent feature $C_A \in \mathbb{R}^{N_a \times F_a}$, where $F_a = 128$ is the number of feature channels.

**Lane Segment Encoder.** Given the map information $\mathcal{M}$, we define the lane node matrix $V_l \in \mathbb{R}^{N_l \times 2}$, where $N_l$ is the number of lane segments, and each entry represents the midpoint between two neighboring nodes along the lane centerline. The connections between these nodes include the predecessor, successor, left, and right neighbors, which are captured by four adjacency matrices $\{\Upsilon_i\}_{i \in \{\text{pre, suc, left, right}\}}$, where each $\Upsilon_i \in \mathbb{R}^{N_l \times N_l}$. For each lane node $v_i \in V_l$, we define its node feature $u_i \in U$ as:

$$u_i = \psi_1(\Delta v_i) + \psi_2(v_i), \quad (13)$$

where $\psi_1$ and $\psi_2$ are MLP blocks, and $\Delta v_i$ denotes the displacement of neighboring lane nodes. The resulting node feature matrix is $U \in \mathbb{R}^{N_l \times F_l}$, where $F_l = 128$ is the number of lane feature channels. We then apply the multi-scale dilated LaneConv operator, as proposed in (Liang et al., 2020), to aggregate the topology features across a larger lane graph:

$$C_L = \sum_\sigma \left( \Upsilon_{\text{pre}}^\sigma U \Theta_{\text{pre}}^\sigma + \Upsilon_{\text{suc}}^\sigma U \Theta_{\text{suc}}^\sigma \right) \\ + \Upsilon_{\text{left}} U \Theta_{\text{left}} + \Upsilon_{\text{right}} U \Theta_{\text{right}} + U \Theta_0, \quad (14)$$

where $\sigma = \{1, 2, 4, 8, 16, 32\}$ represents the dilation sizes, and $\Theta_i$ are the weight matrices associated with the $i$-th connection type. The resulting lane features are represented as $C_L \in \mathbb{R}^{N_l \times F_l}$.

**Drivable Area Encoder.** We construct the drivable area (DA) graph, as in (Zhang et al., 2022), by uniformly sam-

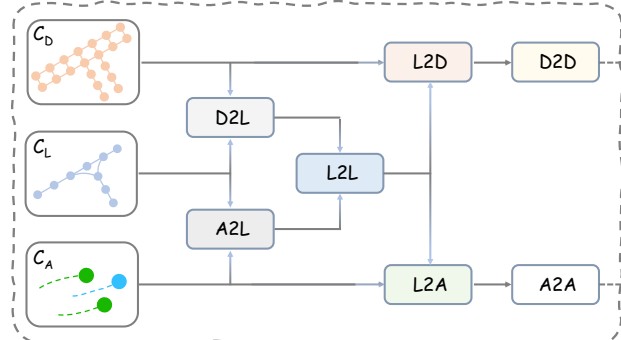

*Figure 6.* Information flow of agent-map feature fusion.

pling drivable nodes centered around the target agent within a 100-meter range and a resolution of 1 meter. These nodes are denoted $V_d \in \mathbb{R}^{N_d \times 2}$, where $N_d$ is the number of drivable nodes. We use eight-neighbor dilated connections to model relationships between the drivable nodes. The features are encoded using a PointNet-like architecture, termed `PointDA`. Specifically, we map surrounding node features to the target node using an MLP block, followed by max-pooling to extract global features. These global features are then concatenated with the target node feature and passed through another MLP block to fuse the features. The final drivable area features are denoted as $C_D \in \mathbb{R}^{N_d \times F_d}$, with $F_d = 32$ being the number of feature channels.

**Feature Fusion.** To capture agent-map interactions, we perform feature fusion on the agent features $C_A$, lane segment features $C_L$, and drivable area features $C_D$, following the approach in (Liang et al., 2020; Zhang et al., 2022). We employ graph attention layers (Vaswani et al., 2017; Liang et al., 2020) to fuse features across different node types: agent-to-lane (A2L), lane-to-agent (L2A), drivable-to-lane (D2L), lane-to-drivable (L2D), and agent-to-agent (A2A). Additionally, we use the `LaneConv` operator for lane-to-lane (L2L), and `PointDA` for drivable-to-drivable (D2D). All attention layers are set to 128 feature channels. The information flow for all fusion processes is shown in Figure 6.

**Feature Adaptor.** To integrate the MaxEnt IRL framework with vectorized features, we design a feature adaptor that transforms the lane-graph features into grid space. We construct a $H_G \times W_G$ grid map centered around the target agent within a 50-meter range, uniformly sampling grid nodes at a resolution of 1.0 meter, which aligns with the drivable area graph. Thus, we have $H_G = W_G = 50$. For each grid node with physical coordinates $(x_i, y_i)$, we assign the corresponding nearest drivable area features $C_D(x_i, y_i)$ as its fused feature. For grid nodes whose distance to the nearest drivable node exceeds 1.0 meter, indicating they are undrivable, we pad their features with zeros. In this way, we complete the adaptation of the vectorized context features

into a grid-based format, which can then be used in the subsequent MaxEnt IRL process.

## A.2. Training Objectives

We detail the mathematical formulations of the training objectives, including both regression and classification losses.

Given $K$ predicted trajectories $\hat{\mathcal{Y}}^k = [\hat{y}_1^k, \hat{y}_2^k, \dots, \hat{y}_{t_f}^k]$ for $k = 1, 2, \dots, K$, and the ground-truth future trajectory $\mathcal{Y}^{\text{GT}} = [y_1^{\text{GT}}, y_2^{\text{GT}}, \dots, y_{t_f}^{\text{GT}}]$, we identify the positive trajectory $k^*$ as the one with the minimum final displacement error relative to $\mathcal{Y}^{\text{GT}}$.

**Regression Loss.** The regression loss is calculated using the Huber loss. Taking the refined trajectory loss $\mathcal{L}_{reg}^{\text{T}}$ as an example, it is defined as:

$$\mathcal{L}_{reg}^{\text{T}} = \frac{1}{t_f} \sum_{i=1}^{t_f} \texttt{Huber}(\|\hat{y}_i^{k^*} - y_i^{\text{GT}}\|_2), \qquad (15)$$

where $\|\cdot\|_2$ represents the $\ell_2$-norm, and the Huber loss function $\texttt{Huber}(\cdot)$ is the smooth $\ell_1$ loss given by:

$$\texttt{Huber}(x) = \begin{cases} 0.5x^2 & \text{if } \|x\|_1 < 1, \\ \|x\|_1 - 0.5 & \text{otherwise,} \end{cases} \qquad (16)$$

where $\|\cdot\|_1$ denotes the $\ell_1$-norm.

In addition, for the trajectory proposal loss $\mathcal{L}_{reg}^{\text{P}}$, we replace $\hat{y}_i^{k^*}$ with the trajectory proposal $\hat{\tilde{y}}_i^{k^*}$. Similarly, for the goal point loss $\mathcal{L}_{reg}^{\text{G}}$, only the final term $\hat{y}_{t_f}^{k^*}$ is considered.

**Classification Loss.** The classification loss $\mathcal{L}_{cls}$ is computed using the Hinge loss (max-margin loss) for binary classification. It is defined as:

$$\mathcal{L}_{cls} = \frac{1}{K-1} \sum_{k \neq k^*} \max(\mathcal{P}_{cls}(\hat{\mathcal{Y}}^k) - \mathcal{P}_{cls}(\hat{\mathcal{Y}}^{k^*}) + \varepsilon, \; 0), \qquad (17)$$

where $\varepsilon$ is the margin, and $\mathcal{P}_{cls}(\hat{\mathcal{Y}}^k)$ represents the probability of trajectory $\hat{\mathcal{Y}}^k$ for $k = 1, \dots, K$.

## B. Additional Quantitative Results

To comprehensively demonstrate the effectiveness of our GoIRL in trajectory prediction, we provide several recent entries from the Argoverse leaderboard, as shown in Table 5, including both supervised and self-supervised models. Our GoIRL achieves a competitive brier-minFDE$_6$ and a superior Brier score relative to strong supervised baselines, such as those with more powerful scene encoders (Zhou et al., 2023) as well as recent self-supervised methods (Lan et al., 2023), highlighting both reliable predictions and strong overall performance. In summary, our approach employs a graph-based IRL framework that effectively mitigates covariate shift while maintaining competitive performance on standard trajectory prediction benchmarks.

*Table 5.* Results on the Argoverse motion forecasting benchmark.

| Method | brier-minFDE$_6$ | Brier score | minFDE$_6$ | minADE$_6$ | MR$_6$ |
|---|---|---|---|---|---|
| LaneRCNN (Zeng et al., 2021) | 2.147 | 0.694 | 1.453 | 0.904 | 0.123 |
| LaneGCN (Liang et al., 2020) | 2.059 | 0.695 | 1.364 | 0.868 | 0.163 |
| AutoBot (Girgis et al., 2022) | 2.057 | 0.685 | 1.372 | 0.876 | 0.164 |
| THOMAS (Gilles et al., 2022a) | 1.974 | 0.535 | 1.439 | 0.942 | 0.104 |
| GOHOME (Gilles et al., 2022b) | 1.983 | 0.533 | 1.450 | 0.943 | 0.105 |
| GO + HOME (Gilles et al., 2021) | 1.860 | 0.568 | 1.292 | 0.890 | 0.085 |
| DSP (Zhang et al., 2022) | 1.858 | 0.639 | 1.219 | 0.819 | 0.130 |
| HiVT (Zhou et al., 2022) | 1.842 | 0.673 | 1.169 | 0.774 | 0.127 |
| MultiPath++ (Varadarajan et al., 2022) | 1.793 | 0.579 | 1.214 | 0.790 | 0.132 |
| GANet (Wang et al., 2023) | 1.790 | 0.629 | 1.161 | 0.806 | 0.118 |
| HeteroGCN (Gao et al., 2023) | 1.751 | 0.591 | 1.160 | 0.789 | 0.117 |
| SIMPL (Zhang et al., 2024) | 1.747 | 0.592 | 1.155 | 0.769 | 0.117 |
| Wayformer (Nayakanti et al., 2022) | 1.741 | 0.579 | 1.162 | 0.768 | 0.119 |
| QCNet (Zhou et al., 2023) | 1.693 | 0.626 | 1.067 | 0.734 | 0.106 |
| SEPT (Lan et al., 2023) | 1.682 | 0.625 | 1.057 | 0.728 | 0.103 |
| **GoIRL (Ours)** | 1.695 | 0.569 | 1.126 | 0.783 | 0.110 |

## C. Additional Qualitative Results

In this section, we provide additional consecutive trajectory prediction results, examples of generalization to drivable area changes, and some representative failure cases.

### C.1. Consecutive Trajectory Prediction

**Experimental Settings.** To effectively demonstrate the prediction performance of our proposed GoIRL model, we directly apply it to the Argoverse tracking dataset (Chang et al., 2019) without any additional training. The experimental settings are fully consistent with those of the Argoverse motion forecasting benchmark. For clarity and relevance, we filter the predictions to include only vehicles within a 50-meter radius of the ego agent and traveling at speeds above 2 m/s, excluding all others from the visualization. We predict six future trajectories for each vehicle, each associated with a probability score. To enhance the visual clarity in the demo video, multimodal trajectories are rendered with varying transparency based on their corresponding probabilities. Specifically, as shown in Figure 7, trajectories with lower probabilities are displayed with greater transparency, ensuring a clean and intuitive visualization.

**Snapshots in the Demo Video.** In consecutive prediction scenarios, we expect the multimodality of predictions to be more evident initially; as the vehicle continues, its intent becomes clearer, leading to reduced diversity in predictions. For example, in straightforward scenarios where vehicles travel straight, the prediction with the highest probability should closely align with the ground truth, while alternative predictions capture slight variations in velocity or endpoint. Conversely, at intersections, significant multimodality typically arises, capturing various potential maneuvers. Some snapshots showcasing these multimodal predictions are highlighted in Figure 7. The visualizations indicate that GoIRL performs robustly in such consecutive predictions, underscoring its effectiveness in practical applications. Additional

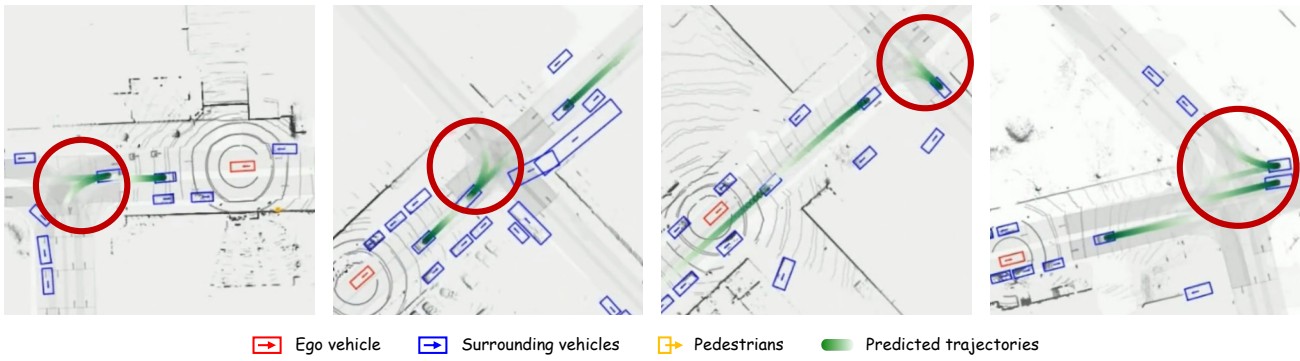

Ego vehicle    Surrounding vehicles    Pedestrians    Predicted trajectories

*Figure 7.* Snapshots with multimodal predictions in the demo video.

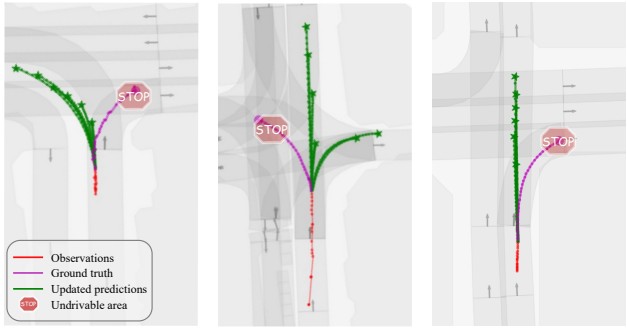

*Figure 8.* Examples of generalization to changes in drivable areas. The forecasted trajectories are in green, and the undrivable area is indicated with a "STOP" sign symbolizing the transformed region.

(a) Poor observation.    (b) Imperfect speed prediction.    (c) Unpredictable lane changing.

*Figure 9.* Visualizations of several representative failure cases.

qualitative results can be found in the latter half of the supplementary video[2] for further evaluation.

## C.2. Generalization to Drivable Area Changes

To further demonstrate GoIRL's capability to address the covariate shift issues, we provide additional qualitative examples, as shown in Figure 8. These case studies illustrate that our GoIRL model can effectively adapt to changes in drivable areas and generate feasible forecasted trajectories, underscoring its strong generalization ability across diverse scenarios involving drivable area changes.

## C.3. Failure Cases

We present several representative failure cases of our model. Figure 9(a) illustrates how poor observations of static or slow-moving agents can negatively impact trajectory predictions. Figure 9(b) demonstrates that inaccuracies in speed prediction can affect longitudinal accuracy. Figure 9(c) highlights a limitation where the model fails to anticipate a lane change in the absence of explicit cues. These examples offer some insights into the conditions under which our model

___
[2]https://youtu.be/MPECgueGRaQ?t=178

may underperform and point to potential directions for enhancing the capability of the IRL-based trajectory predictor.

## D. Discussion: IRL v.s. BC

Inverse reinforcement learning (IRL) and behavior cloning (BC) are two primary paradigms in imitation learning (IL). While BC has been widely applied across various domains, including trajectory prediction, IRL offers several advantages for autonomous driving scenarios. Below, we present an intuitive comparison of these two approaches across three key aspects.

**Understanding Underlying Intentions.** A fundamental difference between BC and IRL lies in their approach to learning behavior. BC directly replicates expert behavior using standard supervised learning without attempting to understand the reasons behind the actions. In contrast, IRL aims to infer the underlying objective or reward function that explains the expert's behavior based on demonstrations (Osa et al., 2018). By understanding the motivations behind actions, IRL allows for better generalization to novel, unseen scenarios, enabling the agent to make decisions that align with the inferred intentions even in different contexts.

**Handling Distribution Shift (Covariate Shift).** In IL, the source domain comprises expert demonstrations, while the target domain represents the learner's reproductions. Since demonstration datasets cannot cover all possible situations, learners often encounter states that were not included in the dataset, leading to out-of-distribution or covariate shift scenarios. BC, which heavily relies on patterns in the training data, struggles to adapt to such unseen states. This limitation often results in compounding errors: small mistakes lead the agent into unfamiliar states, causing further errors and eventual divergence. Conversely, IRL learns a reward function that succinctly encapsulates the desired behavior. This reward function enables the agent to re-optimize its policy dynamically, reducing error accumulation, recovering from unfamiliar states, and adapting more effectively to changes in the environment. Consequently, IRL is more robust in real-world driving conditions, where uncertainty and unpredictability are common, resulting in more reliable trajectory predictions.

**Interpretability.** BC operates as a black-box approach, offering little insight into why certain actions are taken. It also lacks the ability to inherently account for constraints unless they are explicitly represented in the training data. In contrast, IRL provides a learned reward function that offers valuable insights into the decision-making process, enhancing interpretability. Additionally, IRL enables the integration of explicit constraints directly into the reward function. It can also model interactions between multiple agents by incorporating them into the inferred rewards, making it a promising tool for improving the safety and social acceptability for joint multi-agent trajectory prediction tasks. Moreover, by optimizing actions over the inferred reward function, IRL accounts for long-term consequences, allowing the agent to plan trajectories that are optimal over extended horizons. This not only enhances safety and efficiency but also facilitates seamless integration with downstream decision-making and motion planning modules.

In summary, while BC is effective in scenarios with well-represented data, IRL's ability to infer intentions, handle distribution shifts, and provide interpretability makes it a promising alternative for complex trajectory prediction tasks in autonomous driving.

