# OpenReview forum: "GoIRL: Graph-Oriented Inverse Reinforcement Learning for Multimodal Trajectory Prediction"
_ICML.cc/2025/Conference — ICML 2025 poster_

### Official Review · Reviewer_DqsN · 2025-03-08

**Overall Recommendation:** 3

**Summary:**

In this paper, the authors introduce a Graph-oriented Inverse Reinforcement Learning (GoIRL) framework for multimodal trajectory prediction. Specifically, (1) to capture the complex scene context in a structured manner, they use vectorized representations of the environment (scene features), (2) to integrate detailed lane information into the prediction model, they aggregate lane-graph features into a grid space, allowing the model to understand the spatial relationship and constraints of the driving environment, (3) they utilizes a hierarchical parameterized trajectory generator for improving prediction accuracy and an Markov chain Monte carlo (MCMC)-augmented probability fusion for boosting prediction confidence.

## update after rebuttal
I gave a 'weak accept" score because Combining the IRL and graph-based context encoding is a good try in trajectory prediction. Thanks.

**Claims And Evidence:**

Overall, claims made in this paper are supported by evidence.
1. The GoIRL framework integrates MaxEnt IRL with vectorized context representations has corresponding detailed explanation. The Section 3 provides detailed descriptions of the framework and related operations.
2. The claim about the performance “achieves state-of-the-art performance on the large-scale Argoverse & nuScenes” is supported by the quantitative results reported in Table 1, 2, and 3. In these tables, the authors compare their solutions with other existing solutions.

There are several recent works shown in the Argoverse motion forecasting page https://paperswithcode.com/sota/motion-forecasting-on-argoverse-cvpr-2020.
It will be better to compare the proposed solution with some existing solutions rank top on this page.

**Essential References Not Discussed:**

It will be better if the authors could discuss more recent work, for example:

* Lan, Zhiqian, Yuxuan Jiang, Yao Mu, Chen Chen, and Shengbo Eben Li. "Sept: Towards efficient scene representation learning for motion prediction." arXiv preprint arXiv:2309.15289 (2023).
* Zhou, Zikang, Zihao Wen, Jianping Wang, Yung-Hui Li, and Yu-Kai Huang. "Qcnext: A next-generation framework for joint multi-agent trajectory prediction." arXiv preprint arXiv:2306.10508 (2023).
* Yao, Yue, Shengchao Yan, Daniel Goehring, Wolfram Burgard, and Joerg Reichardt. "Improving Out-of-Distribution Generalization of Trajectory Prediction for Autonomous Driving via Polynomial Representations." In 2024 IEEE/RSJ International Conference on Intelligent Robots and Systems (IROS), pp. 488-495. IEEE, 2024.

**Experimental Designs Or Analyses:**

Overall, the experimental designs and analyses make sense. However, as mentioned above, it will be better to compare the proposed solution with more existing solutions to prove the effective of the proposed solution.

**Methods And Evaluation Criteria:**

The proposed methods and evaluation criteria make sense for the trajectory prediction task. It will be better, if the authors could report the accuracy for each prediction step. In this case, readers may get more sense about the performance of the proposed solution, e.g., the model performs better at the first few steps and the performs worse at the last step, etc.

**Other Comments Or Suggestions:**

1. It will be better to report some failure cases, and make some discussions about the failures.
2. The interpretability of the proposed model is also a concern. It will be better to come up with a way to improve the interpretability of model’s predictions.

**Other Strengths And Weaknesses:**

Other Strengths:
1. Combining the IRL and graph-based context encoding is a good try in trajectory prediction. The purpose of the usage of the inference learning is explained clearly and makes sense.
2. The framework design should be generalize well to new data and different domains.

Other Weaknesses:
1. It will be better to report the complexity of the proposed solution. The integration of multiple components make the framework complex, and may involve challenges in real world use case. Reporting the complexity/latency of the solution will be helpful to readers.
2. The performance of the model may rely on the quality of training data. For example, the accuracy of the lane-graph feature, observed trajectories, etc. It will be better to report the performance with noisy inputs.

**Questions For Authors:**

1. Do we need to use different grid space for different scenes/datasets?

**Relation To Broader Scientific Literature:**

The proposed solution combines the Inverse Reinforcement Learning with graph-based context to improve prediction performance and generalization. To achieve this goal, there are several components of this paper were adapted from existing work. For example:
1. Context Encoder:  the Dilated LaneConv operator proposed in LaneGCN is applied.
2. Policy Generator: followed the MaxEnt IRL framework.

**Theoretical Claims:**

This paper does not have challenging mathematical proofs or theoretical claims. The model operations and loss functions used in this paper look reasonable and should work for the trajectory prediction task.

---

> ### Author Rebuttal · Authors · 2025-03-31
>
> We sincerely appreciate the reviewer’s thorough and thoughtful feedback. We are grateful for the recognition of our work’s motivations, technical contributions, framework designs, and experimental results, as well as for the constructive suggestions for improvement. Below, we address each of the reviewer’s comments in detail.
>
> 1. **Accuracy at Each Prediction Step.** Thank you for your valuable suggestion. The following table (https://anonymous.4open.science/api/repo/Anonymous-F687/file/Tab1.png) presents the Displacement Error (DE) per prediction step for the forecasted trajectory. As expected, the model performs more accurately in the initial steps, with errors increasing in later steps. This trend aligns with the inherent uncertainty of long-horizon predictions. We will incorporate this analysis into the revised manuscript.
>
> 2. **Recent Solutions & References.** We greatly appreciate the reviewer’s recommendations for recent solutions. The referenced works propose different paradigms:
>    - Ref. [1] employs a self-supervised approach leveraging all data (including the test set) for pertaining.
>    - Ref. [2] introduces a powerful query-centric paradigm for scene encoding within a supervised learning paradigm.
>    - Ref. [3] focuses on improving out-of-distribution generalization by introducing a novel evaluation protocol.
>
>     Our approach differs in that it emphasizes a graph-based IRL framework to address covariate shift while maintaining competitive performance on standard trajectory prediction benchmarks. We will cite and discuss these works in the revised manuscript.
>
> 3. **Model Inference Latency.** Thank you for this important consideration. Our model achieves an average inference latency of approximately 30 ms on an NVIDIA RTX 3090 GPU using a standard PyTorch implementation. We will include this information in the revision.
>
> 4. **Performance with Noisy Inputs.** We appreciate the suggestion to evaluate our model’s robustness. To assess this, we introduced Gaussian noise with zero mean and varying standard deviations (STD) to the inputs. The results, summarized in the following table (https://anonymous.4open.science/api/repo/Anonymous-F687/file/Tab2.png), indicate that while significant noise degrades prediction accuracy, minor perturbations have minimal impact. This suggests that our model exhibits a degree of robustness to moderate input noise. We will include these findings into the revised manuscript.
>
> 5. **Failure Cases.** Thank you for the valuable suggestion to analyze failure cases. We provide several representative failure scenarios, which can be accessed at https://anonymous.4open.science/api/repo/Anonymous-F687/file/Figure1.jpg.
>    - Figure 1(a) illustrates how poor observations of static or slow-moving agents can negatively impact trajectory predictions.
>    - Figure 1(b) demonstrates that inaccuracies in speed prediction can affect longitudinal accuracy.
>    - Figure 1(c) highlights a limitation where the model fails to predict a lane change in the absence of explicit cues.
>
>     We believe these examples offer some insights into the conditions under which our model may struggle. These failure cases, along with a discussion of their potential causes, will be included in the revised manuscript.
>
> 6. **Model Interpretability.** We sincerely appreciate your insightful comment. Unlike fully supervised models that solely learn from data distributions, our graph-based IRL approach follows the MaxEnt IRL framework, which provides a learned reward function. This offers meaningful insights into the model’s decision-making process and allows for the integration of explicit constraints within the reward function, potentially enhancing interpretability. As discussed in Appendix C, we provide preliminary analysis on this aspect, and we recognize that further exploration of interpretability remains an important avenue for future research.
>
> 7. **Grid Resolution for Different Scenes.** Thank you for this instructive question. The choice of grid resolution involves a trade-off between efficiency and accuracy. In our implementation, we simply use a fixed grid resolution for both the Argoverse and nuScenes datasets, complemented by a refinement module to mitigate its impact. However, adapting the grid resolution to the specific driving scene could further improve performance.
>
> **References**
>
> [1] SEPT: Towards Efficient Scene Representation Learning for Motion Prediction.
>
> [2] QCNeXt: A Next-Generation Framework for Joint Multi-Agent Trajectory Prediction.
>
> [3] Improving Out-of-Distribution Generalization of Trajectory Prediction for Autonomous Driving via Polynomial Representations.

---

> > ### Comment · Reviewer_DqsN · 2025-04-05
> >
> > Thank you for your responses. The authors responded to my questions in a great way. They addressed my concerns. I do not have additional questions. Thank you so much.

---

> > > ### Author Response · Authors · 2025-04-06
> > >
> > > Thank you very much for taking the time to review our work. We sincerely appreciate your encouraging feedback and the opportunity to address your concerns. Your insights have been instrumental in helping us refine and strengthen our manuscript. We are truly grateful for your thoughtful support.

---

### Official Review · Reviewer_ZXUe · 2025-03-13

**Overall Recommendation:** 2

**Summary:**

This paper focuses on the problem of trajectory prediction, which is hard because of the inherent uncertainty and underlying multimodality. Previous method mainly focuses on behavior cloning, which has been shown to have a covariant shift problem. Therefore, this paper proposes to use IRL to solve this problem. In particular, the authors propose a Graph-oriented Inverse Reinforcement Learning (GoIRL) framework, which is an IRL-based predictor equipped with vectorized context representations. They develop a feature adaptor to effectively aggregate lane-graph features into grid space, enabling seamless integration with the maximum entropy IRL paradigm to infer the reward distribution and obtain the policy. Extensive experimental results show that the proposed approach not only achieves state-of-the-art performance on the large-scale Argoverse & nuScenes motion forecasting benchmarks but also exhibits superior generalization abilities compared to existing supervised models.

**Claims And Evidence:**

Yes.

**Essential References Not Discussed:**

No.

**Experimental Designs Or Analyses:**

Yes.

**Methods And Evaluation Criteria:**

Yes.

**Other Comments Or Suggestions:**

No.

**Other Strengths And Weaknesses:**

Strengths:

1.	In general, this paper is well-written and easy to follow. The figures are clear and illustrative to understand.
2.	The idea of using IRL to solve the trajectory prediction task is interesting and promising, it has the potential to avoid covariant shift problems in behavior cloning.

Weaknesses:

1.	Some recent trajectory prediction methods are missing. Although this paper focuses on graph-based representation and IRL, it focuses on a well-studied area that has rapid development and new algorithms. I think one reason is that the most recent work focuses on the waymo dataset, which is larger than argoverse and nuScenes. I think the authors should also include results on the Waymo dataset and compare the SOTA methods in the leaderboard. In particular, I think autoregressive models with tokenized representation have recently become popular and promising and should be considered as baselines.
2.	Not enough qualitative examples are shown in the paper. Figures 4, 5, and 7 only show the multi-modal prediction of a single vehicle scenario. I think the biggest problem in trajectory prediction is learning the interaction between diverse agents. The authors may need to show more examples of the interactive scenarios with multiple agents, together with the comparison with baseline models.
3.	Loss function (12) might be too complex to tune for different datasets.

**Questions For Authors:**

1.	The covariate shift example in Figure 1 is confusing. If the barricade is included in the observation and there are training samples that contain similar situations, I think the predictor will only predict a left turn. If there is no barricade seen in the training sample, then the model fails because of the out-of-distribution problem rather than the covariant shift.
2.	“We employ the target-centric coordinate system, where all context instances and sequences are normalized to the current state of the target agent through translation and rotation operations.” The efficiency of using “target-centric” representation could be low when we need to predict the trajectory of all agents. Can this proposed method be integrated into query-centric representation?
3.	On page 7 line 375, the authors say “Furthermore, we adopt an ensemble strategy (Zhang et al., 2024) to boost prediction performance”. How does this ensemble work? Do methods in Table 3 also use ensemble?

**Relation To Broader Scientific Literature:**

Trajectory prediction is an important task in autonomous driving.

**Theoretical Claims:**

No theoretical claims.

---

> ### Author Rebuttal · Authors · 2025-03-31
>
> We sincerely appreciate the reviewer’s recognition of our research contributions and writing clarity, as well as the constructive feedback and valuable suggestions for improvement. Below, we address the reviewer’s concerns in detail.
>
> 1. **Recent Trajectory Prediction Methods.** Thank you for your thoughtful suggestions. We would like to clarify our choice of datasets. Since our work focuses on empowering IRL with vectorized representations, we selected Argoverse and nuScenes for evaluation based on benchmarking considerations: the graph-based model (DSP [1]) was originally evaluated on Argoverse, while P2T [2], to our knowledge, the only IRL-based trajectory predictor on the public leaderboard, was benchmarked on nuScenes. To ensure fair and meaningful comparisons, we adopted these two datasets as our benchmarks. Nonetheless, we acknowledge the importance of the Waymo dataset and will consider extending our experiments to Waymo in future work. Regarding autoregressive models for trajectory prediction, we include Autobot [3], an autoregressive approach on Argoverse, for comparison. As shown in the table below, our method remains competitive, demonstrating the effectiveness of IRL-based approaches.
> |Method|brier-minFDE$_{6}$|Brier score|minFDE$_{6}$|minADE$_{6}$|MR$_{6}$|
> |:-|:-:|:-:|:-:|:-:|:-:|
> |AutoBot|2.057|0.685|1.372| 0.876|0.164|
> |GoIRL (Ours)|1.796|0.623|1.173|0.809|0.120|
>
> 2. **Qualitative Examples.** We apologize for any confusion regarding our qualitative results. The scenarios presented in Figures 4, 5, and 7 include multiple traffic participants; however, for clarity, we omitted other agents to better highlight the target vehicle’s predictions. A more comprehensive visualization, including all agents, is available in the supplementary video. We will revise the manuscript to explicitly state this and include additional examples in the appendix to better illustrate interactive scenarios with multiple agents.
>
> 3. **Loss Function.** Thank you for your insightful comment. Our loss function follows a standard formulation for two-stage trajectory prediction models, comprising regression losses for both proposal and refined trajectories, along with a classification loss for probability estimation. In practice, we use fixed values of $\alpha=$ 1, $\beta=$ 1, and $\gamma =$ 3 in Eq. (12), which do not require extensive tuning across datasets. We will clarify this in the revised manuscript.
>
> 4. **Covariate Shift Example in Figure 1.** We appreciate the reviewer’s insightful comment and apologize for any confusion. The example in Figure 1 is intended to illustrate a covariate shift scenario in which the drivable area attribute changes. Herein, the barricade represents a constraint indicating that the region behind it is undrivable. In the training dataset, the model learns from observed drivable area annotations and the corresponding labels of ground-truth future trajectories. However, during testing, the drivable attribute is altered in the input while the ground-truth trajectory remains unchanged. This setup aligns with the definition of covariate shift [4], where the input distribution changes while the relationship between inputs and labels remains the same. We will clarify this more explicitly in the revision.
>
> 5. **Query-Centric Representation.** We greatly appreciate the reviewer’s insightful suggestion. Our IRL framework can indeed be integrated into a query-centric representation, and our ongoing work has demonstrated its feasibility in such settings. However, in this paper, our primary focus is on exploring the benefits of a graph-based representation. We will discuss the potential for extending our approach to query-centric representations in the future work section.
>
> 6. **Ensemble Strategy.** Thank you for your thoughtful inquiry. Our ensemble strategy follows a standard approach, where multiple models trained with different random seeds are combined using a weighted k-means algorithm to aggregate the forecasted trajectories. The results presented in Table 3 also incorporate this ensemble strategy, and we will explicitly clarify this point in the revised manuscript.
>
> **References**
>
> [1] Trajectory Prediction with Graph-based Dual-Scale Context Fusion.
>
> [2] Trajectory Forecasts in Unknown Environments Conditioned on Grid-based Plans.
>
> [3] Latent Variable Sequential Set Transformers for Joint Multi-Agent Motion Prediction.
>
> [4] https://en.wikipedia.org/wiki/Domain_adaptation.

---

### Official Review · Reviewer_PdeW · 2025-03-21

**Overall Recommendation:** 2

**Summary:**

GoIRL is a graph-based inverse reinforcement learning framework for predicting multiple possible future trajectories in autonomous driving. It integrates lane-graph features into IRL, uses a hierarchical decoder for accurate predictions, and outperforms supervised models on Argoverse and nuScenes benchmarks.

## Update after rebuttal

In the rebuttal, the authors included the model’s performance on the AV2 leaderboard, enhancing the comprehensiveness of the evaluation. They also presented quantitative results on Drivable Area Changes, which help illustrate the method’s potential for generalization in such scenarios.

However, the performance of GoIRL still lags significantly behind the state-of-the-art method LOF (B FDE 1.63 vs. 1.76). Furthermore, the specific advantage offered by IRL in generalizing to drivable area changes remains unclear. The necessity of addressing this problem through IRL is questionable—alternative approaches, such as prediction models conditioned on the road graph, can also adhere strictly to the drivable area.

Overall, given that the absolute performance is not improved and that the claim of generalizability to drivable area changes seems somewhat ill-posed, I assign a weak reject as my final rating. I recommend conducting additional experiments and comparisons focused on more challenging long-tail generalization scenarios (e.g., drivable area changes, sudden pedestrian emergence) to more convincingly demonstrate the benefits of using IRL. A comparison with SOTA models on such challenging setting would significantly strengthen the paper.

**Claims And Evidence:**

1. GoIRL is the first to integrate the MaxEnt IRL paradigm with vectorized context representation.
This claim appears to be true and highlights the novelty of the proposed approach.
2. GoIRL proposes a hierarchical parameterized trajectory generator and an MCMC-augmented probability fusion method for improved performance.
This claim is well-supported by the ablation study, which clearly demonstrates the performance benefits of each component.
3. It achieves state-of-the-art performance on two large-scale motion forecasting benchmarks.
This claim is well-supported by the quantitative results presented in the paper.
4. It demonstrates superior generalization abilities compared to existing supervised models in handling drivable area changes.
This claim lacks sufficient evidence. The paper provides only a single qualitative example, which is not enough to conclusively support the generalization claim.

**Essential References Not Discussed:**

none

**Experimental Designs Or Analyses:**

Please see the 'Methods And Evaluation Criteria*' part

**Methods And Evaluation Criteria:**

The proposed method leverages Inverse Reinforcement Learning (IRL) to address compounding error and generalizability issues, which is a reasonable and well-motivated approach. Prior work has demonstrated the effectiveness of IRL in similar contexts.

However, the benchmark datasets used in the paper are outdated. Argoverse (2019) and nuScenes (2020) are no longer actively maintained or considered state-of-the-art. More recent and widely adopted benchmarks, such as Argoverse 2 and the Waymo Open Motion Dataset, are not used. As a result, the validity of the claimed improvements is questionable, especially since strong baselines on these newer benchmarks (e.g., MTR, QCNet, LOF) are not included in the comparisons.

Additionally, the evaluation metrics are not comprehensive. The paper claims superior generalization ability, yet provides no quantitative experiments to support this assertion.

**Other Comments Or Suggestions:**

none

**Other Strengths And Weaknesses:**

Please see above

**Questions For Authors:**

none

**Relation To Broader Scientific Literature:**

This paper leverages previous findings in Inverse Reinforcement Learning.

**Theoretical Claims:**

The theoretical claims look good to me in this paper.

---

> ### Author Rebuttal · Authors · 2025-03-31
>
> We sincerely appreciate the reviewer’s recognition of our work’s motivations, technical contributions, and experimental results, as well as the constructive suggestions for improvement. Below, we address the reviewer’s concerns in detail.
>
> ### **1. Choice of Benchmark Datasets & Recent Baselines.**
> We acknowledge the reviewer’s concerns regarding dataset and baseline selection and provide our response as follows:
>
> (1) **Fair Baseline Comparison.** Our choice of Argoverse and nuScenes is primarily motivated by ensuring fair and meaningful comparisons within established benchmarks, as outlined below:
>   - Our supervised graph-based baseline (DSP [1]) was originally evaluated on Argoverse, ensuring a fair and rigorous comparison with our proposed IRL framework.
>   - To the best of our knowledge, nuScenes is the only large-scale benchmark featuring an IRL-based predictor (P2T [2]) on its public leaderboard. Since P2T employs rasterized images as input, evaluating our graph-based IRL model on nuScenes provides a valuable contrast.
>   - Despite being introduced in 2019 and 2020, Argoverse and nuScenes remain widely used, with strong baselines such as LOF [3] and QCNet [4] continuing to benchmark on them. The frequent public submissions further demonstrate their ongoing relevance.
>
> (2) **Comparison to Recent Baselines.** Our model is the first IRL-based predictor with a vectorized representation, which primarily focuses on the comparison against other graph-based approaches and the IRL-based predictor with rasterized representations. As shown in the table below, our GoIRL model also achieves comparable brier-minFDE$_6$ and superior Brier score relative to strong supervised models (e.g., LOF & QCNet), demonstrating reliable predictions and competitive performance. As suggested, we will incorporate these recent baselines in our revised manuscript for a more comprehensive comparison.
> |Method|brier-minFDE$_{6}$|Brier score|
> |:-|:-:|:-:|
> |LOF|1.658|0.626|
> |QCNet|1.693|0.626|
> |GoIRL (Ours)|1.695|0.569|
>
> (3) **Contributions Beyond Benchmarks.** Our work establishes a well-performing IRL-based baseline for trajectory prediction. While we acknowledge the importance of newer datasets, we believe our current evaluation is sufficient and reasonable to validate our contributions. Nevertheless, we plan to explore our method’s potential on additional benchmarks in future work.
>
> ### **2. Evaluation for Generalization in Drivable Area Changes.**
> We appreciate the reviewer’s concern regarding evaluation metrics and have conducted additional quantitative experiments to assess our model’s ability to generalize to drivable area changes. To the best of our knowledge, DSP is the only supervised model that incorporates drivable information. Therefore, we compare our method against DSP by evaluating the success rate across 150 diverse traffic scenarios, each with modified drivable attributes (as described in Section 4.3 of the paper). For each scenario, we randomly alter the drivable attributes near the future ground-truth positions 10 times, yielding 1,500 evaluation cases. The results, summarized in the table below, demonstrate that IRL-based predictors inherently learn interaction dynamics, leading to superior generalization in handling drivable area changes and mitigating domain bias. Moreover, we also provide additional qualitative examples to further illustrate the significance of addressing covariate shift, which can be accessed at https://anonymous.4open.science/api/repo/Anonymous-F687/file/case.jpg. These case studies demonstrate how our GoIRL model effectively handles covariate shifts in terms of drivable area changes in different scenarios.
> |Method|Success Rate|
> |:-|:-:|
> |DSP|15.87%|
> |GoIRL (Ours)|88.13%|
>
> **References**
>
> [1] Trajectory Prediction with Graph-based Dual-Scale Context Fusion.
>
> [2] Trajectory Forecasts in Unknown Environments Conditioned on Grid-based Plans.
>
> [3] FutureNet-LOF: Joint Trajectory Prediction and Lane Occupancy Field Prediction with Future Context Encoding.
>
> [4] Query-Centric Trajectory Prediction.

---

> > ### Comment · Reviewer_PdeW · 2025-04-01
> >
> > The authors have updated their comparison with recent baselines, including LOF and QCNet. While GoIRL demonstrates a strong Brier score, it slightly underperforms these recent baselines in terms of brier-minFDE. Additionally, the paper still lacks evaluation on the AV2 leaderboard, and the rationale for focusing only on predictors within the same category (i.e., graph-based or IRL-based) remains unclear—broader comparisons would help contextualize the model’s overall standing in the field.
> >
> > On the other hand, I appreciate the new quantitative results evaluating generalization under drivable area changes. The significant performance gap between GoIRL and DSP in this setting highlights the strength of the proposed approach in capturing interaction dynamics.
> >
> > Given these improvements, I will raise my score to 2.

---

> > > ### Author Response · Authors · 2025-04-06
> > >
> > > We sincerely appreciate your time and thoughtful feedback. Thank you again for your encouraging comments regarding our quantitative results on generalization under drivable area changes.
> > >
> > > As suggested, we have conducted additional evaluations of GoIRL on the Argoverse 2 leaderboard to facilitate broader comparisons. The results, presented in the table below, indicate that our model consistently achieves competitive performance, particularly in terms of the Brier score, which highlights GoIRL’s strength in producing accurate and reliable trajectory forecasts.
> > > |Method|brier-minFDE$_{6}$|Brier score|
> > > |:-|:-:|:-:|
> > > |LOF|1.63|0.56|
> > > |QCNet|1.78|0.59|
> > > |GoIRL (Ours)|1.76|0.55|
> > >
> > > While GoIRL may slightly underperform recent strong baselines such as LOF on certain metrics like brier-minFDE$_{6}$, we believe further gains could be realized through targeted hyperparameter fine-tuning. Moreover, the primary goal of this work is to explore IRL as a principled and interpretable alternative learning paradigm for trajectory prediction tasks. In this light, GoIRL’s strong generalization ability in handling drivable area changes underscores the potential of IRL-based approaches for robust motion prediction in real-world scenarios.
> > >
> > > Thank you once again for your valuable suggestions. We remain committed to improving performance on standard benchmarks and will continue advancing the capabilities of IRL-based trajectory predictors in future work.

---

### Official Review · Reviewer_igcX · 2025-03-21

**Overall Recommendation:** 2

**Summary:**

The Graph-oriented Inverse Reinforcement Learning (GoIRL) framework is an IRL-based predictor that utilizes vectorized context representations. The author states that the proposed methods overcome the drawbacks of supervised learning techniques. Additionally, a hierarchical parameterized trajectory generator has been incorporated to enhance prediction accuracy. The experiments were conducted on two real-world datasets: Argoverse and nuScenes.

**Claims And Evidence:**

Some claims made in the submission are problematic please refer to the weakness section.

**Essential References Not Discussed:**

Nil

**Experimental Designs Or Analyses:**

The soundness of the experimental designs looks correct.

**Methods And Evaluation Criteria:**

The method evaluation makes sense for the problem of trajectory prediction.

**Other Comments Or Suggestions:**

Nil

**Other Strengths And Weaknesses:**

Strengths
1.	The paper is well-written and easy to follow.
 The experimentation is thorough,
2.	The results on two real-world datasets are promising, outperforming the compared methods.


Weakness

The following are the weaknesses that need to be addressed to improve the manuscript:

Motivation and Claims:

1.	In Figure 1, the authors claim that “During the data collection process, the ground-truth trajectory is labeled as going straight.” However, no evidence is provided to support this claim. The authors should offer clarification or cite a source to validate this statement.

2.	Furthermore, the authors assert that existing supervised models can hardly react to such changes, which is not entirely accurate [1]. There are existing methods that incorporate collision avoidance behavior. For example, if there is a vehicle in front of the ego-vehicle, many models are designed to predict that the ego-vehicle will follow collision avoidance protocols and choose a safer maneuver, especially if such scenarios are present in the training data.

[1] Meng, Dejian, et al. "Vehicle trajectory prediction based predictive collision risk assessment for autonomous driving in highway scenarios." arXiv preprint arXiv:2304.05610 (2023).

3.	In line 071, the author states, “Another critical concern associated with the supervised approach is the modality collapse issue.” This arises because the predictor must generate diverse and plausible predictions, yet many supervised methods tend to produce a single trajectory (deterministic prediction) [2]. Additionally, in exp., the author mentions K plans of 6 and 10; however, K = 1 is not included.

[2] Bae, Inhwan, Young-Jae Park, and Hae-Gon Jeon. "Singulartrajectory: Universal trajectory predictor using diffusion model." Proceedings of the IEEE/CVF Conference on Computer Vision and Pattern Recognition. 2024.

4.	Is “recurrently forecasts control points to represent the trajectory” inspired by Graph-TERN [3]?

[3] Bae, Inhwan, and Hae-Gon Jeon. "A set of control points conditioned pedestrian trajectory prediction." Proceedings of the AAAI Conference on Artificial Intelligence. Vol. 37. No. 5. 2023.



Novelty
The main novelty of the paper lies in the use of MaxEnt IRL (Maximum Entropy Inverse Reinforcement Learning) for the multimodal trajectory prediction task. Existing literature [4,5] already addresses MaxEnt IRL in the context of trajectory prediction. The author should clearly distinguish between these methods as well.

[3] Deo, Nachiket, and Mohan M. Trivedi. "Trajectory forecasts in unknown environments conditioned on grid-based plans." arXiv preprint arXiv:2001.00735 (2020).
[4] T. Hirakawa, T. Yamashita, K. Yoda, T. Tamaki, and H. Fujiyoshi, "Travel Time-dependent Maximum Entropy Inverse Reinforcement Learning for Seabird Trajectory Prediction," in Asian Conference on Computer Vision, pp. 430-435, 2017.
[5]	T. Hirakawa, T. Yamashita, T. Tamaki, H. Fujiyoshi, Y. Umezu, I. Takeuchi, S. Matsumoto, K. Yoda, "Can AI predict animal movements? Filling gaps in animal trajectories using Inverse Reinforcement Learning," Ecoshere, vol 9, no. 10, pp. e02447, 2018.


Experimentation

1.	In Table 4, the addition of the Bezier curve results in only a minor improvement. The authors should explain why the Bezier curve contributes so little to the overall performance.
2.	Additionally, what is the baseline implementation, and which result in Table 4 corresponds to the baseline? This should be clearly stated to help readers understand the comparative performance of the proposed method.

**Questions For Authors:**

Please see Weaknesses

**Relation To Broader Scientific Literature:**

The paper contributes to a real-world application and plays an important role in autonomous driving, intelligent systems, navigation robots, and video surveillance.

**Theoretical Claims:**

Nil

---

> ### Author Rebuttal · Authors · 2025-03-31
>
> We sincerely appreciate the reviewer’s affirmative comments on the experimentation, results, and writing, as well as constructive suggestions and valuable references, which have helped us strengthen our manuscript. Below, we provide detailed responses to each of the concerns.
>
> 1. **Clarification of the Statement in Figure 1.** Thank you for your insightful comments. In fact, the motivating example in Figure 1 is abstracted from a real-world scenario in the Argoverse dataset (please see Figure 5 in the paper), where the original ground-truth trajectory follows a straight path. This example illustrates the necessary adaptations a predictor should make when encountering drivable area changes in the driving environment. If the ground-truth trajectory involved a left turn, we would modify the undrivable region at the left-turn entrance accordingly to reflect such domain bias issues. We will clarify this more explicitly in the revised manuscript.
>
> 2. **Handling Drivable Area Changes.** We appreciate the reviewer’s concern and would like to further clarify our motivation. Our work focuses on handling drivable area attributes, which differ from object-level obstacles (with tracking IDs or bounding boxes). For instance, certain undetectable obstacles, such as regions behind traffic barricades, may appear empty but remain undrivable due to implicit road rules or physical constraints. In practice, drivable area information can be obtained through techniques like occupancy prediction; in our work, we derive it from HD map annotations in the dataset. Our IRL-based method is designed to handle scenarios where drivable attributes differ from those observed during training, and we will clarify this distinction more explicitly in the revised manuscript.
>
> 3. **Deterministic Prediction.** Many thanks for your constructive comment. Our work primarily focuses on multimodal trajectory prediction due to the inherent uncertainty in future intentions. However, we recognize that deterministic prediction remains important. We appreciate the suggested reference (SingularTrajectory [1]) and will include it in our discussion, along with additional experimental results for k=1 in our revised version, as presented in the table below:
> |Dataset|minFDE$_{1}$|minADE$_{1}$|MR$_{1}$|
> |:-|:-:|:-:|:-:|
> |Argoverse|3.365|1.551|0.539|
> |nuScenes|6.530|3.185|0.822|
>
> 4. **Control-Point Based Prediction.** We thank the reviewer for sharing the excellent work, Graph-TERN [2], which presents a similar approach to trajectory representation. We will include it in our references and acknowledge its relevance in the revised manuscript.
>
> 5. **Distinction from Existing IRL-Based Predictors.** We greatly appreciate the reviewer’s valuable suggestion. The key novelty of our approach, compared to prior IRL-based predictors, lies in the representation and processing of driving context. Unlike previous IRL-based methods [3, 4, 5] that rely on rasterized imagery inputs, our method employs a vectorized (graph-based) representation, enabling it to capture geometric and semantic information in traffic scenes more effectively. Moreover, prior works such as [4,5] primarily focus on animal trajectory prediction, where motion dynamics differ significantly from those in human-driven traffic scenarios.
>
> 6. **Contribution of Bézier Curve Parameterization.** We greatly appreciate the reviewer’s detailed feedback. The primary motivation for adopting Bézier curve-based trajectory parameterization is to ensure kinematic feasibility, rather than solely improving trajectory prediction accuracy. While the performance improvement in standard metrics may be marginal, Bézier curves provide smooth, physically plausible trajectories that align with real-world motion constraints. We will clarify this reasoning in the revised manuscript.
>
> 7. **Clarification of Table 4.** Many thanks for your valuable suggestion. Table 4 presents an ablation study quantifying the impact of key components in our trajectory decoder. Each row removes a specific component to isolate its contribution to the overall performance and the last row represents the complete model, incorporating all components. We will make a clear statement in the revised version.
>
> **References**
>
> [1] SingularTrajectory: Universal Trajectory Predictor Using Diffusion Model.
>
> [2] A Set of Control Points Conditioned Pedestrian Trajectory Prediction.
>
> [3] Trajectory Forecasts in Unknown Environments Conditioned on Grid-based Plans.
>
> [4] Travel Time-dependent Maximum Entropy Inverse Reinforcement Learning for Seabird Trajectory Prediction.
>
> [5] Can AI Predict Animal Movements? Filling Gaps in Animal Trajectories using Inverse Reinforcement Learning.

---

### Official Review · Reviewer_LLrG · 2025-03-23

**Overall Recommendation:** 4

**Summary:**

This paper presents Graph-oriented Inverse Reinforcement Learning (GoIRL), a novel IRL-based prediction framework that leverages vectorized context representations. The framework first extracts features from the vectorized inputs and then transforms them into grid space using a feature adaptor to ensure compatibility with existing IRL methods. To improve efficiency and accuracy, GoIRL employs a hierarchical, parameterized trajectory generator augmented with a refinement module that enhances both prediction accuracy and confidence. Extensive experiments demonstrate the effectiveness of the proposed design.

**Claims And Evidence:**

The claims are substantiated by thorough methodological descriptions and comprehensive experimental results.

**Essential References Not Discussed:**

To the best of the reviewer's knowledge, there is no existing work that closely resembles this research.

**Experimental Designs Or Analyses:**

This submission adopts standard evaluation metrics (ADE, FDE, MR) and widely-used datasets (Argoverse 1 and nuScenes) for experimental validation, which is fair and reasonable. However, given the availability of more challenging and diverse datasets such as Argoverse 2 and Waymo, the authors are encouraged to evaluate the proposed method on these benchmarks further to demonstrate its generalization capabilities better.

**Methods And Evaluation Criteria:**

Most recent motion forecasting methods adopt behavior cloning paradigms, which struggle with generalization and domain adaptation. These limitations become especially critical when the environment deviates significantly from the training distribution. In contrast, reinforcement learning (RL) offers a principled solution through its reward-driven learning and interaction-based training, rather than merely fitting to data distributions. Moreover, RL is also well-suited to address the modality collapse issues commonly observed in supervised learning approaches.

**Other Comments Or Suggestions:**

N.A

**Other Strengths And Weaknesses:**

This submission is well-motivated, particularly in addressing the issue of covariate shift. It also provides a well-structured discussion of related work, clearly distinguishing between behavior cloning (BC) and inverse reinforcement learning (IRL) approaches. The proposed method is novel in several aspects: it effectively leverages rich information from vectorized representations and introduces a feature adaptor that enables compatibility with IRL methods. Furthermore, the hierarchical parameterized trajectory generator is designed to mitigate the inefficiencies commonly found in grid-based approaches.

**Questions For Authors:**

The main concern is the omission of several commonly used baselines, such as:

1. HiVT: Hierarchical Vector Transformer for Multi-Agent Motion Prediction
2. GAMoNet: Goal Area Network for Motion Forecasting

Although some of these methods may not be directly comparable—for instance, due to the use of more powerful scene encoders—they should still be included for completeness and to provide a broader context for comparison.

Moreover, given that the proposed method operates in the grid representation space, it would be beneficial to include comparisons with other supervised grid-based approaches to better highlight its advantages:

1. THOMAS: Trajectory Heatmap Output with Learned Multi-Agent Sampling
2. GOHOME: Graph-Oriented Heatmap Output for Future Motion Estimation

Lastly, the paper would benefit from including more examples or case studies illustrating covariate shift, to further emphasize the importance and relevance of addressing this issue.

**Relation To Broader Scientific Literature:**

This paper contributes to the broader scientific literature on motion forecasting by advancing the integration of inverse reinforcement learning (IRL) with vectorized map representations. Traditional approaches in motion forecasting predominantly follow behavior cloning (BC) paradigms, which are known to suffer from issues such as covariate shift and poor generalization to out-of-distribution scenarios. This work addresses these limitations by leveraging the reward-driven and interaction-based nature of IRL, positioning it as a more robust alternative to BC methods.

**Theoretical Claims:**

I have reviewed the IRL algorithms presented in the submission and did not identify any issues.

---

> ### Author Rebuttal · Authors · 2025-03-31
>
> We sincerely appreciate the reviewer’s thorough and professional evaluation of our work. We are grateful for the recognition of our work’s motivations, novelty, contributions, and experimental validation, as well as for the constructive feedback and valuable suggestions for improvement. Below, we provide detailed responses to each of the reviewer’s comments.
>
> 1. **Choice of Datasets.** Thank you for your thoughtful suggestion regarding additional datasets. Our current evaluation focuses on Argoverse and nuScenes, as these datasets have been widely used to benchmark graph-based trajectory prediction methods. Specifically, we selected Argoverse to benchmark against graph-based models and nuScenes to compare against P2T, which, to our knowledge, is the only IRL-based trajectory predictor on the public leaderboard. Ensuring fair and meaningful comparisons was our primary motivation for this choice. Nevertheless, we acknowledge the importance of evaluating on more diverse datasets to better demonstrate the effectiveness of our approach and we will explore extending our experiments to diverse datasets in future work.
>
> 2. **Additional Baselines.** We appreciate the reviewer’s valuable recommendation to include additional baselines for completeness. In the revised manuscript, we will incorporate comparisons with HiVT [1] and GANet [2], as well as grid-based motion prediction methods such as THOMAS [3], HOME [4], and GOHOME [5]. The table below presents the performance comparison:
> |Method|brier-minFDE$_{6}$|Brier score|minFDE$_{6}$|minADE$_{6}$|MR$_{6}$|
> |:-|:-:|:-:|:-:|:-:|:-:|
> |HiVT|1.842|0.673|1.169|0.774|0.127|
> |GANet|1.790|0.629|1.161|0.806|0.118|
> |THOMAS|1.974|0.535|1.439|0.942|0.104|
> |GOHOME|1.983|0.533|1.450|0.943| 0.105|
> |HOME+GOHOME|1.860|0.568|1.292| 0.890|0.085|
> |GoIRL (Ours)|1.796|0.623|1.173|0.809|0.120|
> |GoIRL-Ens (Ours)|1.695|0.569|1.126|0.783|0.110|
>
> 3. **Additional Case Studies on Covariate Shift.**  Thank you for your insightful suggestion. To further illustrate the significance of addressing covariate shift, we provide additional qualitative examples, which can be accessed at https://anonymous.4open.science/api/repo/Anonymous-F687/file/case.jpg. These case studies demonstrate how our GoIRL model effectively handles covariate shifts in terms of drivable area changes in different scenarios.
>
> **References**
>
> [1] HiVT: Hierarchical Vector Transformer for Multi-Agent Motion Prediction.
>
> [2] GANet: Goal Area Network for Motion Forecasting.
>
> [3] THOMAS: Trajectory Heatmap Output with Learned Multi-Agent Sampling.
>
> [4] HOME: Heatmap Output for Future Motion Estimation.
>
> [5] GOHOME: Graph-Oriented Heatmap Output for Future Motion Estimation.

---

### Decision · Program_Chairs · 2025-05-01

**Decision:**

Accept (poster)

**Comment:**

The paper got mixed review. After carefully reading all reviews, the AC believes that the combination of IRL with graph-based representations and grid space is both novel and effective. We agree with reviewers "DqsN" and "LLrG":
"This paper contributes to the broader scientific literature on motion forecasting by advancing the integration of inverse reinforcement learning (IRL) with vectorized map representations". Using a feature adaptor to bridge vectorized maps with grid-based IRL is a an idea worth publishing that helps retain important scene information while enabling efficient prediction. The MaxEnt IRL approach does a good job capturing uncertainty, and the hierarchical decoder with Bézier curves adds smoothness and accuracy to the predicted trajectories.

The results on Argoverse and nuScenes are strong, outperforming several supervised baselines. The generalization to unseen scenarios, like changes in drivable areas, is impressive. Yet, we also agree on the weaknesses raised by other reviewers on the performance on other datasets. We invite the authors to address them for their camera-ready e.g., "While GoIRL may slightly underperform recent strong baselines such as LOF on certain metrics like brier-minFDE, we believe further gains could be realized through targeted hyperparameter fine-tuning."